# Convergent genome evolution shaped the emergence of terrestrial animals

Jialin Wei[1], Davide Pisani[1,2], Philip C. J. Donoghue[2], Marta Álvarez-Presas[3,4,5] & Jordi Paps[1,5 ✉]

The challenges associated with the transition of life from water to land are profound[1], yet they have been met in many distinct animal lineages[2–5]. These constitute a series of independent evolutionary experiments from which we can decipher the role of contingency versus convergence in the adaptation of animal genomes. Here we compare 154 genomes from 21 animal phyla and their outgroups to reconstruct the protein-coding content of the ancestral genomes linked to 11 animal terrestrialization events, and to produce a timescale of terrestrialization. We uncover distinct patterns of gene gain and loss underlying each transition to land, but similar biological functions emerged recurrently pointing to specific adaptations as key to life on land. We show that semi-terrestrial species evolved convergent functional patterns, in contrast with fully terrestrial lineages that followed different paths to land. Our timeline supports three temporal windows of land colonization by animals during the last 487 million years, each associated with specific ecological contexts. Although each lineage exhibits distinct adaptations, there is strong evidence of convergent genome evolution across the animal kingdom suggesting that, in large part, adaptation to life on land is predictable, linking genes to ecosystems.

The colonization of land by life is one of the most notable transitions in the history of Earth, substantially shaping modern ecosystems, life forms and the planet itself[1]. Terrestrialization has occurred multiple times independently within the animal kingdom, including among arthropods[5], vertebrates[4], rotifers[2], molluscs[6], annelids[2], nematodes[2], tardigrades[7] and onychophorans[3]. Nevertheless, each of these natural experiments in terrestrialization had to overcome similar physiological and environmental challenges[2]. Phenotypic adaptations—such as water-retentive skin or cuticle[2], adapted immune systems[8], changes in skeletal design and locomotion[2], elevated metabolic rates[9], developmental adaptations (such as encapsulated larvae and brooding) and adaptation of vision in aerial environments[10]—show widespread convergence across terrestrial lineages, suggesting largely predictable responses to similar environmental pressures. At the genotype level, recent studies have shown that genomic changes, including gene innovation[11], duplication[12] and loss[13], were crucial to major metazoan evolutionary transitions. Furthermore, specific genes (for example, aquaporin-coding genes) have been linked to terrestrialization in several clades[14], and genomic changes have been associated with individual lineages[15–18]; these support the roles of genes related to metabolism, stress response, osmoregulation and immunity in terrestrialization. However, in comparison to land plants[19], the genomic basis of terrestrialization across animal lineages remains largely uncharacterized. These parallel natural experiments provide a unique opportunity to determine whether terrestrialization has led to lineage-specific contingent genomic adaptations or whether these are predictable changes in response to the same environmental challenges.

Here we apply a comparative genomics pipeline to a dataset of 154 genomes to explore the role of convergence and contingency in the evolutionary response of animal genomes to the process of terrestrialization, and to establish the timeline of acclimatization of animals to land. Our results reveal that independent terrestrial events were driven by the emergence of similar biological functions, although semi-terrestrial and fully terrestrial lineages exhibit different patterns of genomic adaptation to terrestrialization. We identify three temporal windows in which animals colonized land, offering a new chronological framework for these transitions.

## Genome dynamics in terrestrialization

We designed an approach that we named intersection framework for convergent evolution (InterEvo), which identifies the intersection of biological functions between different sets of genes that were independently gained or reduced in different nodes along the phylogeny (Extended Data Fig. 1 and Methods). In brief, we mined 154 genomes—151 from 21 animal phyla and 3 from non-animal holozoans (Fig. 1, Extended Data Fig. 2 and Supplementary Table 1)—and filtered them by completeness. These represent the diversity of animals and our sampling focuses on species flanking nodes that represent terrestrialization events. In this dataset, we identified 11 such events[20] (Fig. 1): node 1 bdelloid rotifers, node 2 clitellate annelids, node 3 Stylommatophora (land gastropods), node 4 nematodes (roundworms), node 5 tardigrades (water bears), node 6 onychophorans (velvet worms), node 7 arachnids, node 8 myriapods (centipedes and millipedes), node 9 *Armadillidium* (woodlice), node 10

[1]Bristol Palaeobiology Group, School of Biological Sciences, University of Bristol, Bristol, UK. [2]Bristol Palaeobiology Group, School of Earth Sciences, University of Bristol, Bristol, UK. [3]Institut de Biologia Evolutiva (CSIC-Universitat Pompeu Fabra), Barcelona, Spain. [4]Present address: Departament de Biologia Evolutiva, Ecologia i Ciències Ambientals, Facultat de Biologia & Institut de Recerca de la Biodiversitat (IRBio), Universitat de Barcelona, Barcelona, Spain. [5]These authors contributed equally: Marta Álvarez-Presas, Jordi Paps. ✉e-mail: jordi.paps@bristol.ac.uk

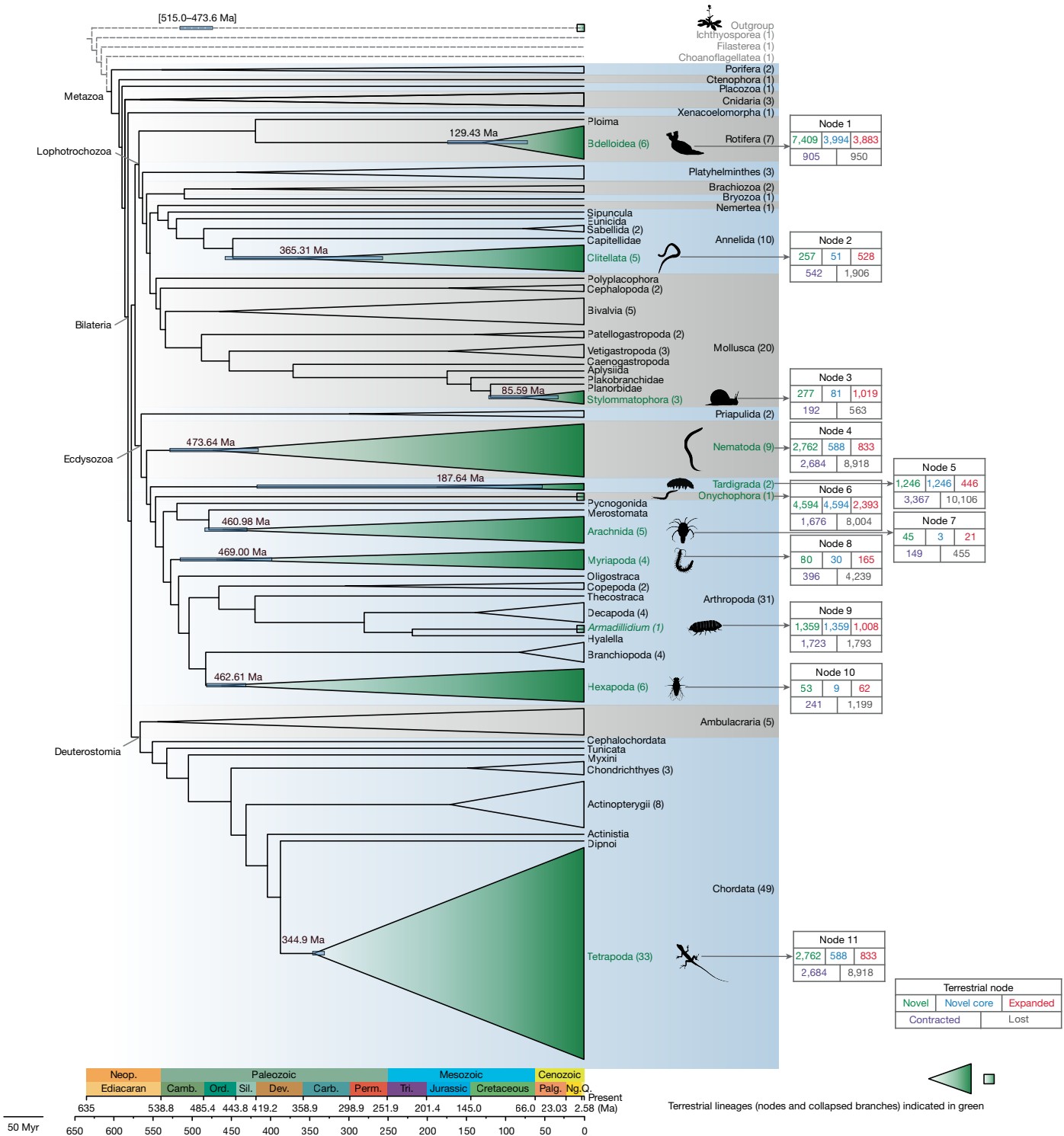

**Fig. 1 | Reconstruction of ancestral genomes and timescale across animal terrestrialization.** The phylogeny is based on 154 sampled taxa, with taxon sampling numbers shown after clade names in parentheses. Terrestrial events are highlighted in green text. HG content of terrestrial nodes is displayed in each corresponding box (from left to right, top to bottom: novel HGs, novel core HGs, expanded HGs, contracted HGs and lost HGs). Branch lengths are scaled to evolutionary time, with divergence times and 95% highest posterior density intervals indicated for terrestrial nodes. Organism silhouettes (sourced from phylopic.org and the authors, released under public domain dedication (CC0 1.0 Universal)) represent the 11 terrestrial events. Geological time abbreviations are as follows: Neop., Neoproterozoic (part); Camb., Cambrian; Ord., Ordovician; Sil., Silurian; Dev., Devonian; Carb., Carboniferous; Perm., Permian; Tri., Triassic; Palg., Paleogene; Ng., Neogene; Q., Quaternary. Myr, million years.

Hexapoda (insects and allies) and node 11 tetrapods (land vertebrates). For Onychophora, Tardigrada and Malacostraca (*Armadillidium*), only one or two genomes were available at the time of starting these analyses.

The 3,934,362 protein sequences from these genomes were clustered into 483,458 homology groups (HGs), groups of proteins that

have distinctly diverged from other groups, comprising orthologues and/or paralogues[21]. Using a previously described approach[11,13,19], we reconstructed the HG content for the key nodes in the tree and classified the HGs based on their mode of evolution: gene gains (novel, novel core and expanded) and gene reductions (contracted and lost).

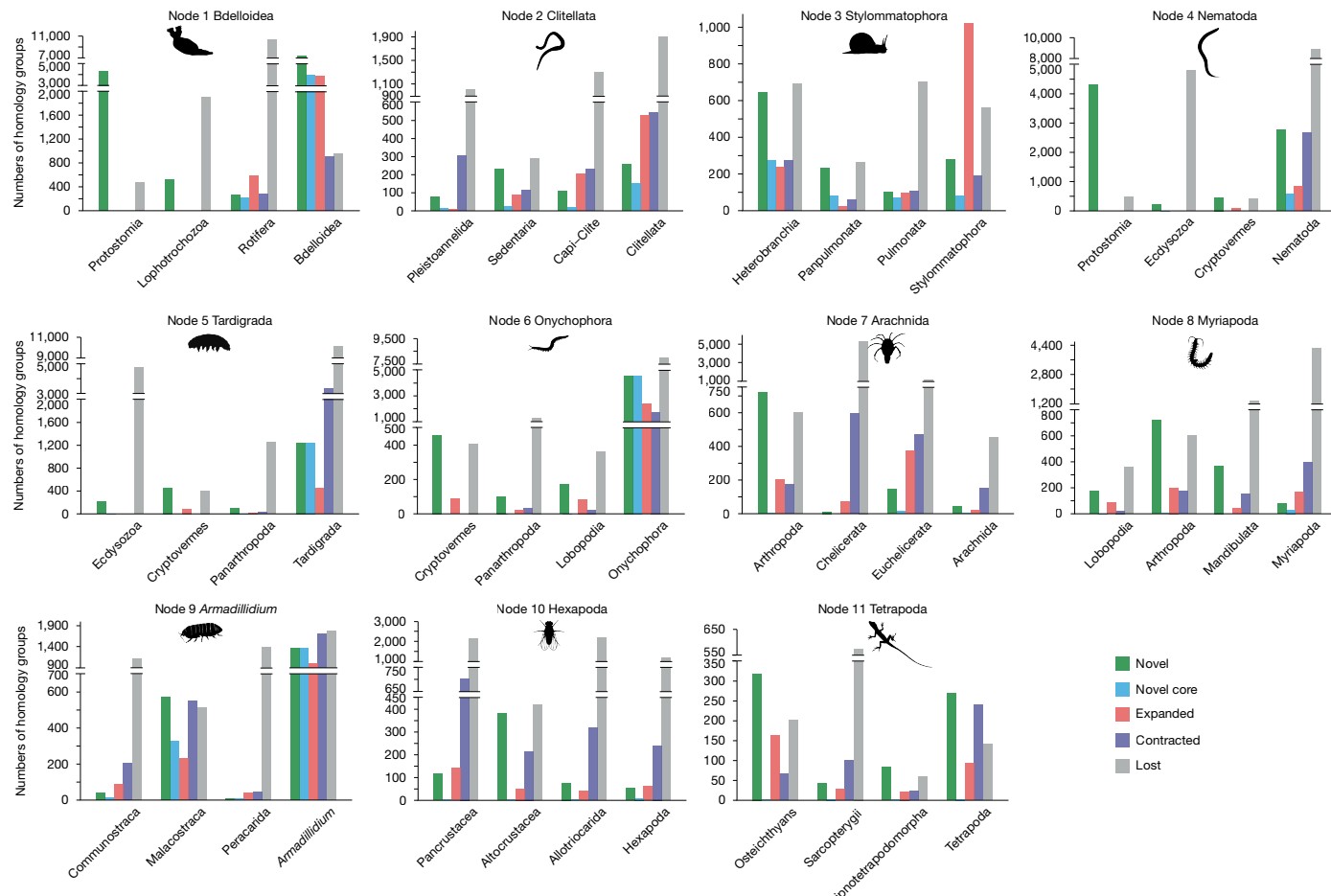

**Fig. 2 | Comparison of HG content across terrestrial nodes and their ancestors.** A total of 154 genomes were analysed to infer HGs and reconstruct ancestral states. Each bar chart represents one terrestrial event node and its three immediate ancestors. For each node, five categories of HGs are quantified: novel, novel core, expanded, contracted and lost. The *y* axis indicates the number of HGs in each category of each clade. Organism silhouettes (sourced from phylopic.org and the authors, released under public domain dedication (CC0 1.0 Universal)) represent 11 terrestrial events.

Novel HGs are those that are present in the ingroup and absent in all the outgroups, with core HGs being present in all the species of the ingroup (permitting one absence). Lost HGs are those that are absent in the ingroup but present in the sister groups and other species in outgroup. Finally, we used CAFE5[22] to infer expanded or contracted HGs, which are those presenting an increase or reduction, respectively, in the number of gene copies (more detailed definitions are provided in Methods and Supplementary Table 2).

Terrestrialization nodes are characterized by a large turnover of gene gains and reductions (Fig. 2). All terrestrial lineages display large gene gains (novel genes and expansions) compared to their immediate ancestors, except arachnids, hexapods and the novel genes in myriapods. Novelty is much higher in bdelloid rotifers, nematodes, tetrapods and land gastropods, the latter only in gene expansions, a finding supported by a previous study[15]. Similarly, terrestrial gene reduction (losses and contractions) is pervasive except in arachnids and hexapods; there is no increase in gene loss in land gastropods, tetrapods and bdelloid rotifers, the latter probably due to the massive losses in the last common ancestor (LCA) of the sampled rotiferans. Nematoda, Tardigrada and Onychophora show the largest gene losses, in agreement with previous studies[12,13], together with Rotifera. Although sparse taxon sampling and the fast-evolving nature of some lineages can inflate gene turnover estimates[13,23], the pattern persists after normalization by divergence time, measured as the accumulation of novel and novel core HGs per million years (Supplementary Table 3). Expansion and contraction need no correction as the birth–death model is intrinsically scaled by branch length. For the significance of gene turnover, a permutation test confirmed that the observed novel gene rates found in terrestrial lineages are significantly higher than in aquatic nodes (*P* = 0.0015; Extended Data Fig. 3a and Supplementary Information section 1.1.1). In summary, most terrestrialization events seem to display high levels of gene turnover, reflecting genome plasticity[24] during the animal transition from water to land associated with new environmental challenges. Arachnids and hexapods show lower levels of plasticity, which may indicate that their evolution was dominated by gene co-option instead.

## Convergent functions via gene gains

To infer functional convergence across the 11 terrestrial events, we annotated the functions of their novel and novel core HGs using both gene ontology (GO)[25,26] and Pfam protein domains[27]. The number of shared GO terms among at least ten nodes are shown in Fig. 3a and shared Pfams among at least five nodes are shown in Fig. 3c. For novel HGs, there are 118 GO terms shared by different combinations of at least 10 nodes (green bars in Fig. 3a; Supplementary Table 4 and Supplementary Fig. 1), and 26 for novel core HGs (blue bars in Fig. 3a, and Supplementary Table 4). Our analyses show that novel gene families that emerged independently in different terrestrialization events are involved in osmosis (regulation of water transport in cells), metabolism (namely fatty acids, probably related to changes in diet), reproduction, detoxification, sensory reception and reaction to stimuli.

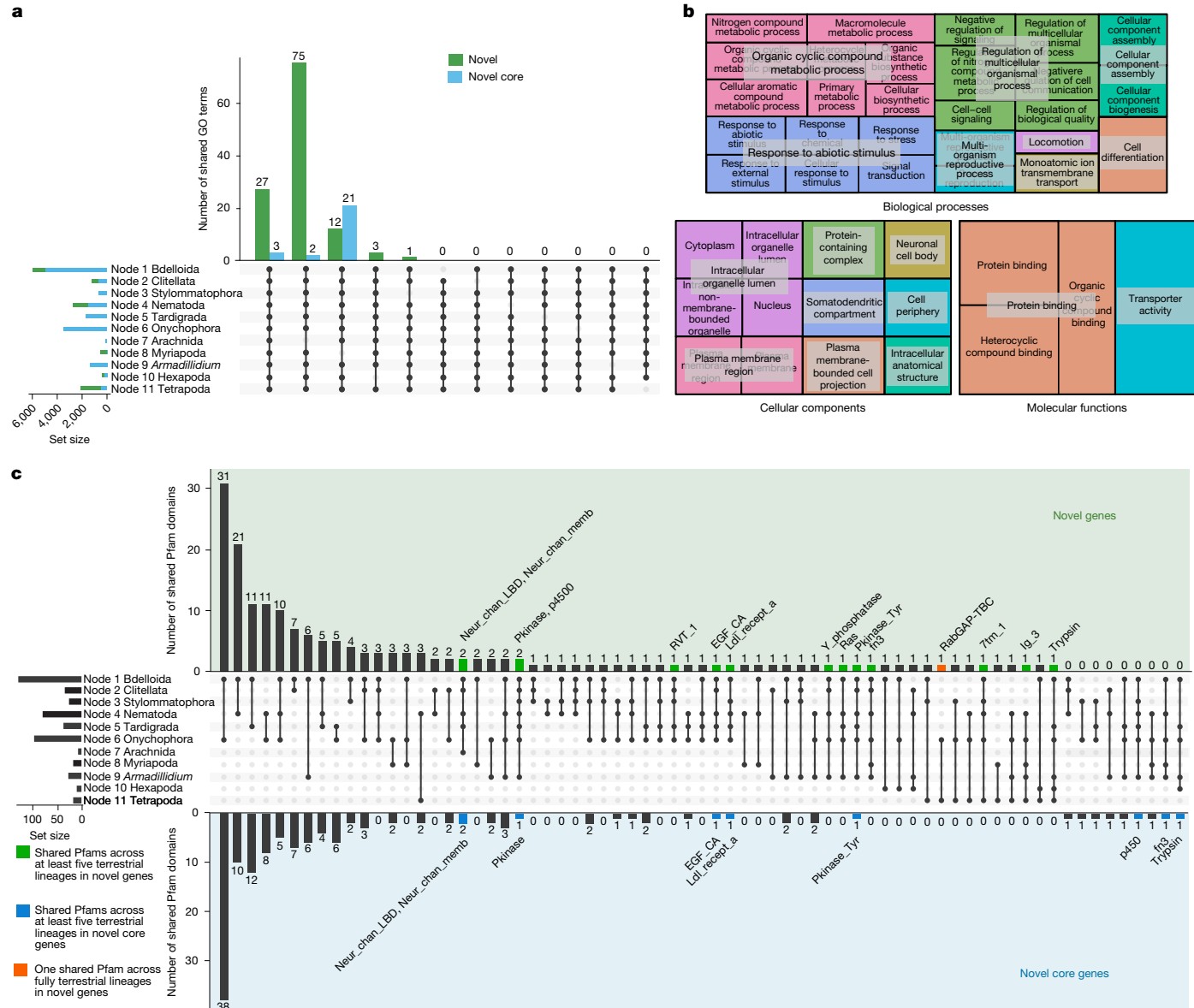

**Fig. 3 | Convergent functional landscape of gene novelty in animal terrestrialization. a**, Distribution of shared GO terms of gene novelty across terrestrialization. Bars indicate the number of GO terms from novel HGs and novel core HGs shared by at least ten terrestrial nodes. The UpSet diagram was generated using the UpSetR R package[60]. **b**, Tree map visualization of GO terms from novel HGs shared across terrestrialization. The 55 most specific GO terms from the 118 shared novel HGs are grouped into 3 major GO categories: biological processes, cellular components, and molecular functions. Tree maps were generated with REVIGO[61]. The hierarchical layout illustrates the relationships among GO terms within each category. **c**, Distribution of shared Pfam domains of gene novelty across terrestrialization. Bars indicate the number of Pfam domains from novel HGs (green) and novel core HGs (blue) shared by at least five terrestrial nodes, the orange bar indicates only one Pfam domain shared among fully terrestrial lineages. These Pfams are labelled above the bars using short names.

All gene families reported here stem from the genome-wide comparative analysis of every HG, and none was chosen a priori.

The 55 'most specific' ('bottom' in the GO hierarchy) GO functions (Fig. 3b) in novel HGs include locomotion, membrane ion transport and transporter activity (osmosis), response to stimulus and neuronal functions (detection and reaction to stimulus), as well as metabolic, reproductive and developmental processes (lifecycle and diet adaptations). Additionally, cellular components include plasma membrane (related to better nutrient uptake, cell barriers and detoxification in adaptation to life on land[28]) and protein-containing complex (a crucial factor of membrane protein insertion[29]). Pfam domains echo these functions, recovering osmoregulation by neurotransmitter-gated ion channel domains, stimulus and neuronal functions by transmembrane receptor, and detoxification by cytochrome P450 (Fig. 3c).

Moreover, the total number of HGs performing these functions also increased in terrestrial nodes (Supplementary Information section 1.2.1 and Extended Data Fig. 4), especially in Bdelloidea, Clitellata, Tardigrada, Onychophora, *Armadillidium* and Tetrapoda. Genes encompassed by these GO terms in humans (tetrapods) and the fruit fly (hexapods) also highlight the importance of biological functions linked to terrestrialization (Extended Data Table 1 and Supplementary Information section 1.2.2). Biological functions specific to terrestrial nodes further support these key adaptations to survival in terrestrial environments. Unique GO or unique Pfams are GO terms or Pfams associated with novel genes that are present in terrestrial nodes that are absent in the GO terms or Pfams of their ancestor nodes (Supplementary Figs. 2 and 3). All nodes except bdelloid rotifers and arachnids contain unique GOs, and all except stylommatophorans and arachnids

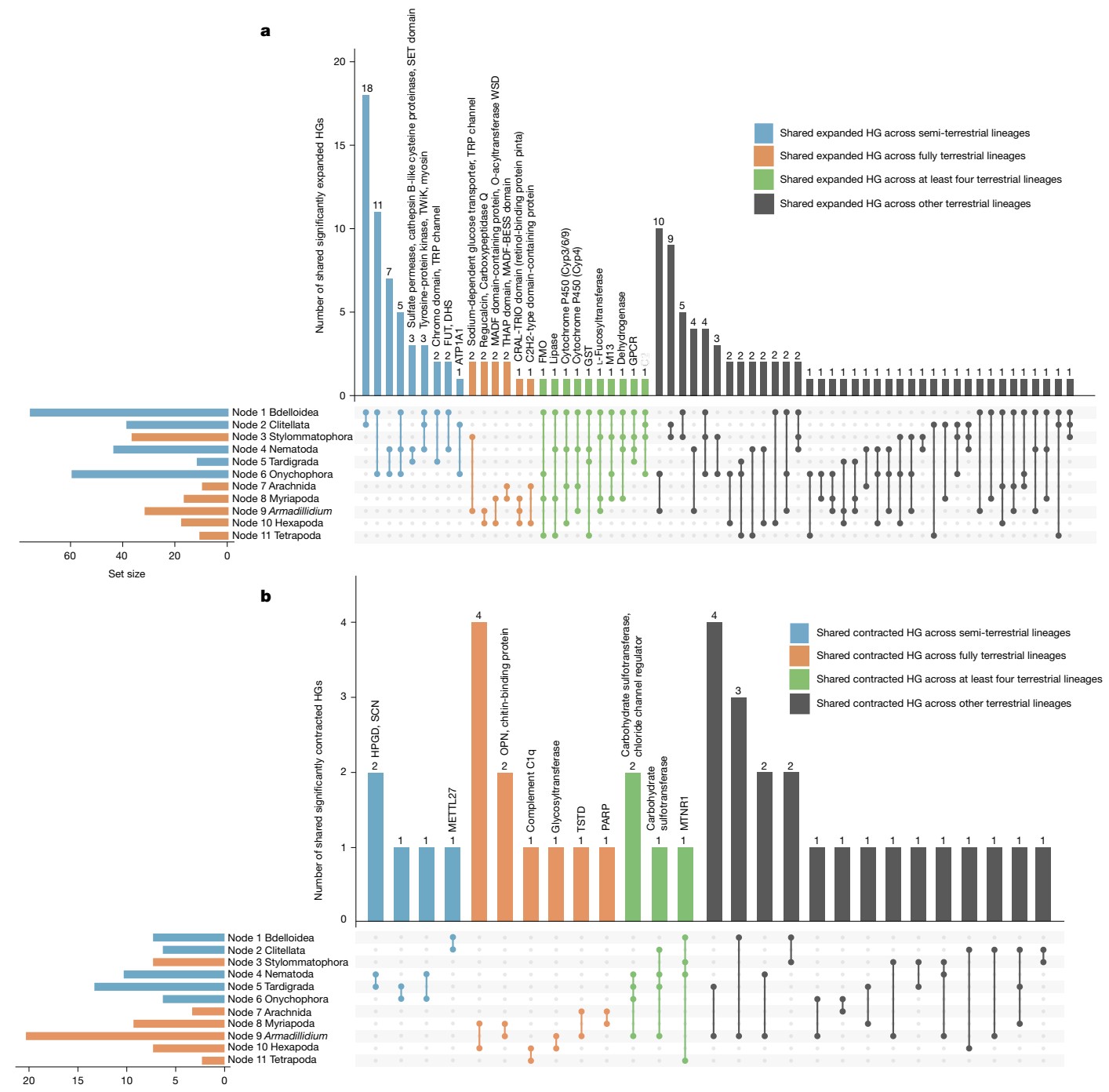

**Fig. 4 | Convergent patterns of shared expanded and contracted HGs across animal terrestrialization. a**, Shared expanded HGs across terrestrial nodes. **b**, Shared contracted HGs across terrestrial nodes. The UpSet diagrams show the intersections of HGs across different combinations of 11 terrestrial nodes. Bars indicate shared expanded and contracted HGs among at least four terrestrial events (green), semi-terrestrial events (blue), and fully terrestrial events (orange). Semi-terrestrial and fully terrestrial groups are differentiated by blue and orange in set sizes (bottom left). HGs with no more than three members are labelled above the bars, other HGs are listed in Supplementary Table 7 (expanded) and Supplementary Table 10 (contracted). Approaches for HGs expansion and contraction inference and functional annotations are described in the Methods.

contain unique Pfams. Shared unique GOs and Pfams are related to metabolism (fatty acid metabolism and kinase activity) and ion transport, which would have helped terrestrial animals maintain water and osmotic balance[30] and allow them to interact with new terrestrial environments, diets and adapt their life cycles. Although some functions of terrestrial novel genes represent exaptations from freshwater ancestors (Supplementary Fig. 4 and Supplementary Table 5), these novel genes remain functionally distinct from genes in aquatic lineages (Extended Data Fig. 3b and Supplementary Information section 1.1.2). Also, some

functions appear to be contingent, being gained early and later lost in terrestrial events (Supplementary Fig. 5 and Supplementary Table 6).

For gene families predating these transitions, we investigated the presence of convergent gene expansions and contractions across the terrestrialization events using CAFE5[22]. These gene families are shared with aquatic ancestors and relatives, so their convergent change in gene repertoire is an indication of parallel exaptations. Our analysis revealed ten HGs that significantly expanded their gene copy numbers in different combinations of four terrestrial nodes (Fig. 4a and Supplementary

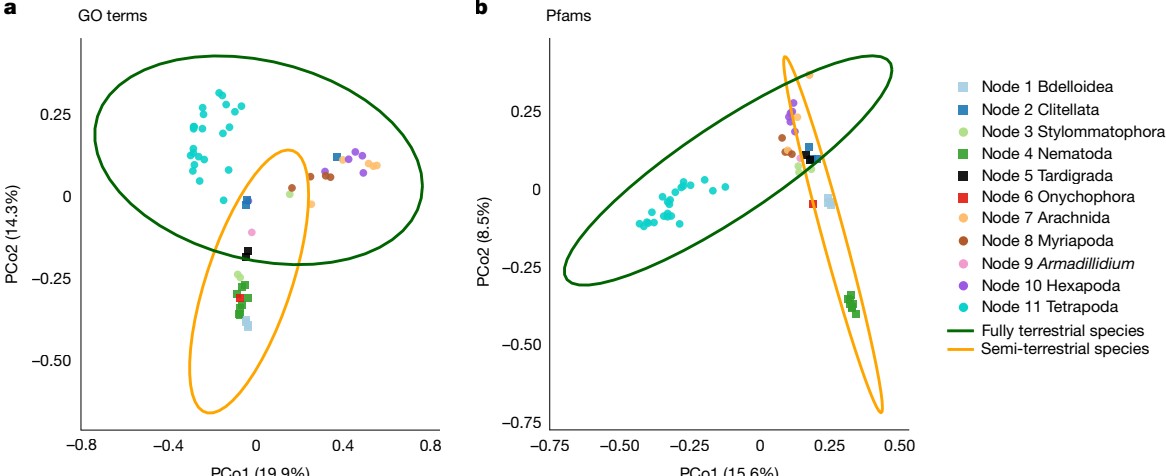

**Fig. 5 | PCoA of GO terms and Pfam domains associated with novel genes in semi- and fully terrestrial species. a**, PCoA of Jaccard dissimilarities based on GO terms presence/absence profiles. **b**, PCoA of Jaccard dissimilarities based on Pfams presence/absence profiles. Each dot represents 1 of the 61 sampled terrestrial species, coloured by taxonomic group as indicated in the legend. Distances between dots correspond to Jaccard dissimilarities. Statistical ellipses highlight the semi-terrestrial group (orange; including Bdelloidea, Clitellata, Nematoda, Tardigrada and Onychophora) and the fully terrestrial group (green; including Stylommatophora, Arachnida, Myriapoda, *Armadillidium*, Hexapoda and Tetrapoda). Ellipses were generated using normal distribution parameters to visualize clustering patterns of taxonomic groups (95% confidence). The two axes represent the first two principal coordinates (PCoA1 and PCoA2) with their respective explained percentages of variation. Group separation was tested with PERMANOVA, which showed the significant differences between semi- and fully terrestrial groups for GO terms ($R^2 = 0.0995$, $P < 0.01$) and Pfam domains ($R^2 = 0.0992$, $P < 0.01$). Group dispersions did not differ (GOs: $P = 0.128$; Pfams: $P = 0.064$). Approaches for functional annotations are described in the Methods.

Table 7); no shared expanded HGs were found in more than four events. These convergently expanded gene families are involved in detoxification, oxidative stress, metabolism and reception of stimuli. Notable examples include the gene families cytochrome P450, which has crucial roles in xenobiotic metabolism particularly in the digestive tract[31]; flavin-containing monooxygenases, essential for processing toxic plant metabolites[32]; and glutathione *S*-transferase, which reduces reactive oxygen species and shows contraction in cetaceans[33]. We also find expansions in the G-protein-coupled receptor family, which is crucial for sensing environmental stimuli such as odours and light[34]. GO term enrichment analysis of expanded HGs, using the bilaterian ancestral genes as background, further supported the importance of stimulus response and ion transport functions in terrestrialization (Supplementary Information section 1.2.3, Supplementary Fig. 6 and Supplementary Table 8).

In summary, our results suggest that gene gains (novel, novel core and expanded gene families) are a key driver across all the transitions from water to land, indicating that functions such as response to stimuli, oxidative stress, lipid metabolism and ion transporter activity had an important role in these adaptive processes.

## Gene reduction marks land adaptation

Gene reduction is another important genetic change in terrestrial events, including lost and contracted genes. Lost HGs are relatively higher in number than gene gains in most nodes (Fig. 2). We identified lost HGs shared in terrestrial events (Supplementary Fig. 7 and Supplementary Table 9) and found the Dbl-homology domain gene family lost in 8 out of 11 terrestrial events, and the pleckstrin-homology domain gene family lost in 7 out of 11 terrestrial events; these are retained mainly in bdelloids, stylommatophorans, myriapods and tetrapods. For reference, there are no gene families convergently expanded or contracted in more than four nodes (Fig. 4), thus finding lost HGs in seven or eight terrestrial nodes is remarkable. Both domains are components of guanine nucleotide exchange factors of Rho GTPases (RhoGEF), implicated in regeneration (neurons[35] and muscles[36]) and wound healing[37]. Other lost HGs among terrestrial events include

chlorophyllase protein family (chlorophyll degradation[38]), potentially indicating dietary shifts during land colonization, and the Shugoshin C-terminal domain-containing protein, which regulates chromosome cohesion and segregation during meiosis and is involved in reproduction[39].

Gene families showing convergent reduction in copy number also point to key adaptations to life on land. There are four HGs that are convergently contracted in at least four terrestrial lineages (green bars in Fig. 4b, and Supplementary Table 10): chloride channel protein members (osmoregulation[40]), two different carbohydrate sulfotransferases (extracellular communication and adhesion[41]), and melatonin-related receptors (circadian rhythms[42]).

## Semi versus fully terrestrial lineages

We categorized the terrestrial lineages as semi-terrestrial or fully terrestrial according to their dependence on water, as no consensus definition of terrestriality is universally accepted. Semi-terrestrial animals rely on humid environments to avoid drying out, and include bdelloid rotifers, nematodes, tardigrades and some microscopic annelids, which require a film of water or pore spaces to live in[43], as well as onychophorans and other annelids (such as clitellates). Fully terrestrial animals are less water-dependent, and encompass lineages such as land gastropods[44], Arachnida, Myriapoda, *Armadillidium* (woodlice), Hexapoda[5] and Tetrapoda. We compared GO and Pfam compositions associated with novel genes of terrestrial animal clades to capture function variation, performing both principal component analysis (PCA; Supplementary Fig. 8) and principal coordinates analysis (PCoA; Fig. 5 and Supplementary Information section 1.3). In the PCoA based on Jaccard dissimilarity, semi-terrestrial and fully terrestrial groups showed partial separation, and permutational multivariate analysis of variance (PERMANOVA) confirmed significant differences between the two groups with both GO terms (Fig. 5a; $R^2 = 0.0995$, $P < 0.01$) and Pfams (Fig. 5b; $R^2 = 0.0992$, $P < 0.01$); group dispersions did not differ (GOs: $P = 0.128$; Pfams: $P = 0.064$). The enriched novel gene functions reveal that semi-terrestrial species carry an expansive and versatile toolkit for environmental flexibility, emphasizing cuticle remodelling,

visual development and stress response, while fully terrestrial species display a small and streamlined set centred on neuronal development and ion membrane homeostasis vital for permanent colonization (Supplementary Table 11).

Notably, consistent with this expansive-versus-streamlined enrichment pattern, semi-terrestrial groups share broad biological functions whereas fully terrestrial lineages show little overlap. Gene gains (Supplementary Information sections 1.3.2 and 1.3.3 and Supplementary Fig. 9 for novel genes, Fig. 4a and Supplementary Table 7 for expanded genes) across semi-terrestrial lineages converged on crucial functions for land adaptation, including circulatory system development, osmoregulation, nutrient processing, muscle function, energy metabolism, detoxification and sensory response mechanisms. These adaptations enabled essential physiological processes required for semi-terrestrial animals to cope with soil-dependent environments, from basic survival needs such as gas exchange, locomotion and nutrient uptake to environmental challenges such as osmotic stress and exposure to pollutants. By contrast, fully terrestrial lineages show limited convergence in the functions associated with gene novelty, with no shared GO terms and only one Pfam domain among novel genes and few shared expanded HGs (Figs. 3c and 4a, Supplementary Fig. 9 and Supplementary Information section 1.3.4). Most shared adaptations among fully terrestrial lineages are found in arthropods, where each terrestrial lineage emerged independently from aquatic ancestors that likely started from a similar genetic toolkit and evolved parallel streamlining later. Only glucose transport and stimulus sensing mechanisms are shared between woodlice and land snails, suggesting that fully terrestrial lineages probably evolved through diverse rather than common adaptive patterns. In addition, both semi-terrestrial and fully terrestrial lineages display diverse gene reduction patterns, with few shared reductions (Fig. 4b for contraction, Supplementary Fig. 7 and Supplementary Table 14 for lost genes) except within arthropods, indicating low convergence of gene reduction across both habitat categories.

## Unique adaptations in terrestrial events

By uncovering novel core and exclusively expanded HGs, we inspected the gene functions associated with each of the 11 terrestrial nodes (Supplementary Information section 1.4 and Supplementary Tables 15–18). These include stress-response genes in bdelloid rotifers, nervous system and muscle adaptations of clitellates, shell formation, mucus secretion and estivation genes in land snails, and cuticle-related genes in nematodes. Tardigrades exhibit unique stress-resistance genes, whereas onychophorans share traits such as oxygen adaptation and nutrient uptake with woodlice. Arthropods and tetrapods, as well-studied terrestrial lineages, were further explored for their distinct adaptations.

Arthropods, the most diverse animal phylum, originated in the sea and colonized the land multiple times independently. In this study, we focus on the fully terrestrial clades Hexapoda, Myriapoda and Arachnida and the crustacean *Armadillidium*[5] (Supplementary Information section 1.4.7). These lineages exhibit convergent evolution of traits for terrestrial adaptation, such as exoskeleton structure, water conservation and sensory development (Fig. 4a and Supplementary Table 7). For instance, myriapods and hexapods expand gene families linked to the synthesis of the exoskeleton wax layer responsible for waterproofing[45]. Similarly, retinol-binding protein genes required in the retinal pigment cells expanded in arthropods to adapt their vision to light conditions on land[46]. Hexapods[5] show enriched GO annotations in expanded genes (Supplementary Table 15) related to moulting (for example, terpenoid metabolic process, juvenile hormone metabolic process, sesquiterpenoid metabolic process and steroid metabolic process) and vision (for example, rhodopsin biosynthetic process).

The other major lineage of terrestrial animals is land vertebrates (Supplementary Information section 1.4.8), which show both novel and expanded genes with enriched GO annotations related to immunity functions (Supplementary Table 18 for novel HGs, Supplementary Tables 15 and 16 for expanded HGs): T cell co-stimulation, positive regulation of activated T cell proliferation, and innate immunity-related processes (for example, neutrophil degranulation and specific granule lumen). Similar innate immunity functions are also found in expanded gene families, such as the Ly-6/uPAR family, siglecs, mucins and resistin. Previous studies have also supported innate immunity as crucial to evolving a specialized and reinforced epidermis with an active keratinization process and a resistant outer stratum corneum[47]. These defend against pathogens that spread in the terrestrial environment, forming both physical and chemical barriers[48], as evidenced by our study.

## Temporal windows of terrestrialization

The invasion of land by life influenced global biogeochemical cycles through effects on carbon storage and weathering, representing a notable milestone in the evolutionary history of the planet[1]. Land plant colonization paved the way to new habitats for animals and fungi and produced the emergence of new ecosystems[49]. Molecular timescales estimate the age of lineages, including soft-bodied animals that may not be well-represented in the fossil record. Here we focus on the terrestrialization events for which we have more than one taxon or genome. Our molecular evolutionary timescale (Fig. 1) is congruent with other recent studies[50] and shows that the animal conquest of land occurred in three major temporal windows. These windows might not overlap and may be separated by millions of years, each contributing to the complexity of terrestrial ecosystems.

The first temporal window of terrestrialization occurred between the Middle Cambrian and Middle Ordovician epochs. Early land plants emerged[49] approximately 515.0–473.6 million years ago (Ma), quickly followed by nematodes and arthropods. Our study suggests that nematodes (533.9–421.9 Ma), myriapods (521.9–402.8 Ma), hexapods (487.6–436.9 Ma) and arachnids (489.7–435.2 Ma) were among the first animals to transition to land, overlapping the rise of early land plants; an arthropod-focused study similarly reports temporal concordance[5]. A study estimated the origin of nematodes between Ediacaran and Silurian periods[51] (620–455 Ma), overlapping, or even preceding, our interval. These early terrestrial species developed traits helping mitigate desiccation and providing structural support, including the arthropod exoskeletons and the nematode cuticles. In our analyses, gene gains in these lineages shared functions associated with cuticle formation, exoskeleton maintenance and lipid metabolism, as well as involvement in responses to drought, excessive light and oxidative stress (Supplementary Table 19), consistent with selection for water conservation and stress tolerance in patchy, intermittently wet terrestrial settings shaped by cryptogamic and bryophyte covers[52].

The second temporal window of terrestrialization spans the Late Devonian to Early Carboniferous subperiod, a time of episodic flooding, deepening soils and strongly seasonal wetlands[53]. In this ecological setting, clitellate annelids (464.5–262.8 Ma) and the first tetrapods (351.2–337.7 Ma) independently adapted to land. The first land vertebrates evolved limbs for locomotion, lungs for aerial respiration and skin barriers to minimize water loss[4]. Clitellates adapted their nervous and muscular systems to cope with terrestrial challenges, enhancing locomotion and desiccation resistance. These species contributed to the establishment of modern terrestrial niches by enhancing nutrient cycling, improving soil structure and influencing ecosystem communities, thereby laying the conditions for further evolutionary innovations in terrestrial life. The floodplains of this period likely provided the ecological opportunities and selection pressures that drove these terrestrial transitions.

The third temporal window of terrestrialization, between 130–86 Ma during the Cretaceous period, saw bdelloid rotifers (180.9–78.4 Ma) and land gastropods (127.1–39.3 Ma) making their way onto land and

sharing it with dinosaurs, as well as early mammals and birds. Bdelloid rotifers evolved exceptional stress tolerance mechanisms, including resistance to desiccation, extreme temperatures, and radiation, enabling them to thrive in harsh environments. Meanwhile, terrestrial snails developed adaptations such as shell formation, mucus secretion, and estivation to withstand diverse climatic conditions. At the molecular level, both clades exhibit gene expansions (Supplementary Table 19) in HGs, including ammonium transporters for water and ion homeostasis, NADP-dependent oxidoreductases[54] and G-protein-coupled receptors for stress resistance. These shared adaptations are likely to reflect Cretaceous greenhouse landscapes, characterized by high sea levels, angiosperm expansion, coastal wetlands and seasonally dry microhabitats[55] that favoured water and ion conservation and broad stress tolerance.

## Discussion

We applied comprehensive comparative genomic analysis to uncover the convergent evolutionary processes underlying 11 independent terrestrialization events. Our results reveal that these are marked by extensive gene turnover across all 11 lineages, adaptations to terrestrial environments. We found that terrestrial events generally display a high level of genomic novelties, mainly related to osmoregulation, stress response, immunity, sensory reception, metabolism and reproduction. We found shared ion transport functions in gene novelty, supporting their critical role in the adaptation from water to land by maintaining water and ionic balance. This is especially crucial for animals adapting to low-salinity and dry environments, as it involves changes in osmoregulation to maintain ion and water homeostasis and prevent water loss. Gene reductions show notably large numbers in terrestrial lineages, with some convergent losses of genes related to regeneration[56]. Although all terrestrial lineages exhibit a certain degree of convergent functional evolution, semi-terrestrial lineages share functional and molecular features, whereas fully terrestrial animals do not. However, it should be noted that habitat classifications are diverse and not universal. For example, another classification, including cryptic forms, poikilohydric organisms and homoiohydric organisms, categorizes Myriapoda and woodlice as cryptic forms[2], whereas here they are classified as fully terrestrial[5]. More comparisons using various classifications are needed in the future. In addition to the convergent emergence of biological functions, each terrestrial lineage exhibits unique adaptations to land. The distinct features indicate the various genomic pathways to thrive in terrestrial environments. Additionally, three major temporal windows of terrestrialization identified by our molecular timescale occurred during the Ordovician, Devonian–Carboniferous and Cretaceous periods. These windows were potentially driven by major ecological and geological changes, forming new terrestrial niches. These results largely support the temporal congruence between the rise of land plants and the first window of terrestrial animals, providing new insights into the tempo of terrestrialization. There are interesting convergences between the adaptation to life on land by plants and terrestrial animals[19]. Plants also evolved genes linked to adaptations to life outside of the water similarly to animals, such as lignin to avoid desiccation and environmental responses (abscisic acid, salicylic acid and jasmonic acid). However, they also present new genes related to UV light protection, a signal that is not observed in animals.

The study faces certain limitations, such as the classification of terrestrialization highlighted above. Moreover, annotating lost and contracted genes poses challenges as these are lost in the most common model organisms, which are the reference for many functional annotations. For example, a significantly contracted HG in nematodes contains no gene copies in *Caenorhabditis elegans* and only a few poorly annotated copies in other nematodes. Functional analysis for such HGs often relies on distant homologues in humans or fruit flies, which may not accurately reflect functions owing to substantial sequence divergence

across lineages. In more extreme cases, where the HGs are lost in traditional model organisms, annotation becomes virtually impossible. Similarly, many lost genes are classified as uncharacterized proteins, reflecting their absence in well-studied terrestrial models. Another methodological limitation is that we did not determine whether gene duplications occurred at the terrestrial nodes or independently within lineages, as CAFE5 infers gene expansions based on copy number, not gene trees. However, the observed expansions remain robust and meaningful as they consistently occur in terrestrial lineages, regardless of when they arose. Also, although our study relies on a robust phylogenetic framework, phylogenetic position incongruence complicates interpretations of terrestrial transitions, such as the debated relationships with Chelicerata. We followed recent studies that place Xiphosura as the sister group to Arachnida[57], implying a single origin of terrestrialization in arachnids, whereas some placements propose that Xiphosura may be nested within arachnids[58], which would suggest an alternative scenario. Additionally, there is limited taxon sampling for certain lineages, such as tardigrades, onychophorans and woodlice, which may lead to HG numbers that are not representative of the gene content of the clade. In the future, with more and more genomes being sequenced, the inclusion of more taxa in datasets will be possible. Future efforts should also focus on developing advanced annotation tools for lost and contracted genes, such as machine learning approaches (for example, language models[59]) to overcome challenges caused by sequence divergence and limited homologues. Moreover, improving gene family expansion inference by integrating gene tree-based approaches will be crucial to pinpoint duplication events more precisely.

This study uses an integrative approach, seamlessly combining comparative genomic analysis, functional annotation and evolutionary timescale reconstruction. By leveraging the InterEvo framework, which assesses the overlap of biological functions in genes repeatedly gained or reduced in these transitions, we systematically reveal the convergent genomic patterns across diverse metazoan taxa, capturing the breadth of animal diversity and providing a robust methodology for studying convergent genome evolution. Furthermore, the analysis incorporates diverse comparisons across terrestrial lineages, offering a comprehensive perspective on both global trends and category-specific patterns of terrestrial adaptations. Many genomic adaptations to terrestrial animal life are convergent, suggesting broadly predictable molecular responses. Yet, convergence is part of the story. Each terrestrial lineage also displays its own contingent adaptations, shaped by its unique evolutionary history, genomic background and ecological context. Even when facing similar challenges, different lineages often arrive at distinct molecular solutions, reflecting their ancestral constraints and trajectories. Terrestrialization, therefore, illustrates the interplay between convergence and contingency, highlighting both the repeatability and the uniqueness of evolutionary innovation.

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

## Methods

### Taxon sampling and HGs inference

We compiled 154 genome samplings from published projects uploaded in UniProt[62], NCBI[63], Ensembl[64] and other resources (Supplementary Table 1). 154 genomes were downloaded, containing 3,934,362 predicted proteins, including 151 metazoan and 3 unicellular organism genomes. A side script provided by OrthoFinder (primary_transcript.py) and Cd-hit v.4.8.1 (ref. 65) (using a similarity threshold of 1.00) were used to extract the canonical proteins from the original data. The quality of canonical proteins from 154 genomes were assessed by BUSCO v.5.4.7 (ref. 66) (Supplementary Table 1). The completeness greater than 85% and fragmentation less than 15% were preferable. It should be noted that we considered not only genome completeness but also the habitat and phylogeny of the species, so we selected some genomes that did not perfectly meet the above standard. We then inferred HGs by Orthofinder v.2.5.5 (ref. 67), using dependencies of MAFFT v.7.505 (ref. 68) and DIAMOND v.2.1.8 (ref. 69). The HGs were then used for analysing gene content.

### Guide tree

We drew a guide tree that was used for gene expansions/contractions analyses and the later time tree. This guide tree, based on the species positions inferred from previous literature[70–73], was used to build the phylogeny of metazoans through the following steps. While some nodes are controversial (namely the position of sponges, ctenophors, or acoels), they are far removed from the terrestrial nodes of interest. First, we started from the conserved single-copy genes in the Metazoa_odb10 from BUSCO v.5.4.7 (ref. 66). *Homo sapiens* contains 943 of these genes, which were extracted to serve as reference orthologous. These identified conserved protein sequences were aligned using MAFFT v.7.505 (ref. 68) and trimmed using trimAl v.1.4.rev.15 (ref. 74) to remove poorly aligned regions. Next, we concatenated the trimmed alignments into a single supermatrix using FASconCAT-G v.1.05.1 (ref. 75). Finally, the concatenated supermatrix was used to build the phylogeny with IQ-TREE v.2.2.2.6 (ref. 76), using C60 + G + I model, using the guide tree as a constraint, and performing 1,000 bootstrap replicates. The resulting phylogeny, with branch lengths representing genetic changes, was subsequently used in CAFE5 for further analysis (see following methods).

### Gene content analysis

Novel HGs: HGs that are present in at least one species within the LCA of a lineage (following we called a node), while being absent in all species of outgroup.

Novel core HGs: HGs that are present in all species within a node (or absent only once for node containing more than three species), while being absent in all species of outgroup. For the node with two species, novel HGs are equal to novel core HGs.

Lost HGs: HGs that are lost in all species within a node, while being present in the sister groups and other species in outgroup.

Expanded HGs: the increase in the number of gene copies occurred within HGs, often due to gene duplication events.

Contracted HGs: the reduction in the number of gene copies occurred within HGs.

Ancestral HGs: all HGs present in a node.

Novel, novel core and lost HGs were inferred by our host pipeline Phylogenetically Aware Parsing Script described by Paps and Holland[11] (GitHub: https://github.com/PapsLab) with Perl v.5.30.0.

Expanded and contracted HGs were inferred by CAFE5[22]. First, we generated an ultrametric phylogenetic tree with ape, TreeTools and phytools packages in R, based on phylogenetic tree built using IQ-TREE. CAFE5 was launched with the ultrametric tree. Owing to the large dataset, we were unable to run the entire phylogeny at once; therefore, we split the phylogeny into three smaller trees: Lophotrochozoa, Ecdysozoa and Deuterostomia. For each smaller tree, CAFE5 was run with Poisson distribution and error model, applying two- and three-lambda models ten times each to test convergence of Model Base Final Likelihood (-lnL). We selected the highest lnL from two- and three-lambda models to compare their fit using a likelihood ratio test with chi-squared distribution (via lmtest package in R), which indicated that three-lambda models are a better fit for all three phylogenies ($P < 0.001$). However, further tests using simulation function of CAFE5 revealed that values (including lambda and -lnL) of three-lambda model of Deuterostomia fluctuated, while that of two-lambda model were stable, thus the two-lambda model was judged to be the better fit for this phylogeny. For Lophotrochozoa and Ecdysozoa, the simulation tests showed stable values for the three-lambda models, which were therefore chosen as the better fit (Supplementary Table 20).

### Novel core HG validation

To test the robustness, novel core HGs were tested by BLASTp v.2.14.0 +[77] using NCBI RefSeq database[78] (downloaded on 23 August 2023), which contains a broad range of high-quality molecular sequences. We launched BLASTp locally and searched novel core HGs against RefSeq records, excluding protein sequences from the in-groups (terrestrial nodes) with the option "-negative_taxidlist". The results shown that BLASTp returned very weak hits; the vast majority of sequences had e-value > $10^{-10}$ and identity <50% (Supplementary Table 21).

### Permutation test analysis

**Novel HGs gain rate.** We evaluated if the number of novel genes emerging per million years in terrestrial nodes differ from aquatic nodes. We collected the rate of emergence of novel gene in 11 terrestrial nodes and randomly selected 11 aquatic nodes (Actinopterygii, Ambulacraria, Bivalvia, Branchiopoda, Chondrichthyes, Cnidaria, Decapoda, Platyhelminthes, Priapulida, Sabellida and Vetigastropoda). We calculated the observed total evolutionary rate in the 11 terrestrial nodes as the total number of novel HGs divided by total divergence time ($R_{terr} = 4.900$). We then performed 10,000 bootstrap draws: in each permutation we sampled (with replacement) 11 aquatic nodes from this pool, recalculated the evolutionary rate ($R_{boot}$ = total novel HG counts divided by total divergence time) and recorded the value, producing a null distribution of novel gene rates in aquatic nodes. The empirical one-tailed $P$ value was the proportion of bootstraps with $R_{boot} \geq R_{terr}$.

**Functional repertoire.** We assessed if the GO term composition of terrestrial lineages differs from that of aquatic lineages. We included lineages with the biggest taxon sampling from random aquatic lineages, including Actinopterygii, Ambulacraria, Bivalvia, Branchiopoda, Cnidaria, Decapoda and Platyhelminthes. We converted the GO matrix derived from the novel genes for each lineage into a binary presence/absence matrix, then quantified the dissimilarity between terrestrial and aquatic GO term profiles by measuring the proportion of non-shared terms (Jaccard distance). For the permutation test, we randomly reshuffled the 'aquatic/terrestrial' labels across lineages 10,000 times; in each permutation we rebuilt the two group profiles and recalculated Jaccard distance between them. The empirical $P$ value was the proportion of permutations distance ≥ the observed distance.

Both analyses were conducted in R using the packages vegan, car and ggplot2.

### Functional annotation and enrichment analysis

For each terrestrial event, we selected one species as representative. *Rotaria sordida* in Bdelloidea, *Eisenia andrei* in Clitellata, *Candidula unifasciata* in Stylommatophora, *Pristionchus pacificus* in Nematoda, *Ramazzottius varieornatusa* in Tardigrada, *Epiperipatus broadwayi* in Onychophora, *Centruroides sculpturatus* in Arachnida, *Rhysida immarginata* in Myriapoda, *Armadillidium nasatum* in *Armadillidium*, *Drosophila melanogaster* in Hexapoda, *H. sapiens* in Tetrapoda.

For annotating Pfam domains and GO terms of the HGs of interest, egg-NOG-mapper v.2 (ref. 79) was applied online with default parameters. Further analysis of names of genes of interest was performed in UniProt[62] using mapping IDs in sequence headers. We also used PANTHER 19.0 (ref. 80) to classify genes with 'protein class'.

We conducted GO enrichment analysis to find overrepresented GO terms in novel and expanded HGs of terrestrial events, using background of GO terms hitting all HGs present in the LCA of Bilateria. Using Fisher's exact test, we compared the number of HGs hitting each GO term between the terrestrial events and bilaterian background. The $P$ values for multiple comparisons were corrected with the Benjamini–Hochberg method. GO terms with adjusted $P$ values < 0.05 were considered significantly enriched. In another way, we also used all HGs present in the LCA of terrestrial events as background to perform enrichment analysis, such as GO terms hitting expanded HGs of hexapods comparing with GO terms hitting all HGs present in LCA hexapods. However, to ensure normalization across all terrestrial events, we chose bilaterian background for the following analysis.

To identify biological functions driving the separation of semi-terrestrial and fully terrestrial groups (following the PCoA), we tested differential presence of GO terms or Pfams between semi-terrestrial and fully terrestrial groups using binary matrices (present/absent). Functional terms that lacked variability (present in all species or in none) were discarded. For every remaining feature we compiled a 2 × 2 contingency table (presence/absence number of species in two habitat categories) and subjected it to a two-tailed Fisher's Exact Test in R, using the marginal totals across entire pool of species as the background. $P$ values were corrected for multiple comparisons using the Benjamini–Hochberg method. The functional terms with adjusted $P$ < 0.05 were considered significantly enriched and those of $P$ < 0.01 were reported. To retain biological relevance, we excluded the functional terms present in ≤10% proportion in both groups.

## PCoA and PCA

To compare the distribution in GO terms linked to novel and ancestral HGs among semi-terrestrial and fully terrestrial lineages, we performed a PCA. PCA was conducted using the prcomp function in R. The GO terms of species were plotted using the first two principal components, PC1 and PC2. Statistical analyses were applied to assess differences between semi-terrestrial and fully terrestrial groups. Analysis of variance (ANOVA) and Tukey's honest significant difference (HSD) test was performed on the principal components scores to evaluate significant differences among these two groups and pairwise comparisons. Then, a multivariate analysis of variance (MANOVA) was conducted to examine the combined effect of these two groups on PC1 and PC2, respectively. Two ellipses, using normal distribution-based ellipse fitting, were merged based on their habitats, representing semi-terrestrial and fully terrestrial groups. The semi-terrestrial group includes bdelloid rotifers, clitellates, nematodes, tardigrades and onychophorans, while fully terrestrial group includes stylommatophorans, arachnids, myriapods, *Armadillidium*, hexapods and tetrapods. The analyses were performed in R using the ggplot2, ggforce and car packages.

However, because shared absences might bias Euclidean-based PCA on binary presence/absence data, inflating similarity between groups that simply lack many of the same features, we further performed PCoA. We quantified compositional differences in GO term and Pfam presence/absence profiles between semi-terrestrial and fully terrestrial species based on Jaccard dissimilarity. Pairwise dissimilarities among species were computed using the Jaccard distance in vegan R package. We then performed PCoA on the Jaccard distance matrix. The two axis labels explain the percentages of Jaccard distance variation. To test for overall group difference between semi-terrestrial and fully terrestrial groups, we performed a PERMANOVA (adonis2 function) on the Jaccard distances with 10,000 permutations, with reassigning species to the two groups. To confirm the PERMANOVA results are not

driven by unequal within-group spread, we tested for homogeneity of multivariate dispersion using betadisper function with permutations of sample–group labels (centroids recomputed each 999 permutation). Plots were generated using ggplot2 package and group ellipses represent 95% concentration regions. The comparison between freshwater and terrestrial species used the same methods described above.

## Molecular clock

Molecular clock analysis was performed using a two-step approach in MCMCTree[81] (PAML package[82]). Using the previously described concatenated alignment of 943 conserved orthologous genes generated using BUSCO v.5.4.7 (ref. 66), MAFFT v.7.505 (ref. 68), trimAl v.1.4.rev.15 (ref. 74) and FASconCAT-G v.1.05.1 (ref. 75). The analysis was conducted in two steps. In the first step, branch lengths were estimated by maximum likelihood using CODEML[83], which calculated the gradient and Hessian of the likelihood function at the maximum likelihood estimates. We used the Empirical + F model (model = 3) and an independent rates clock model (clock = 2). Subsequently, MCMCTree was executed to estimate divergence times, using the same independent rates clock model and discrete gamma distribution with 4 categories and a shape parameter alpha = 0.5. Using an R script provided by MCMCTree tutorial (GitHub: https://github.com/sabifo4/Tutorial_MCMCtree), the prior for the substitution rate was determined based on the approximate root age (591.255 Ma), resulting in a gamma distribution with shape $\alpha$ = 2 and scale $\beta$ = 5.1. For each analysis, we ran the MCMC for about 20 million generations, with the first 100,000 generations discarded as burn-in, sampling every 1,000 generations to obtain 20,000 samples. To ensure convergence and reliability of the results, we performed six independent Markov chain Monte Carlo runs. We assessed convergence using Tracer v.1.7.2 (ref. 84), which showed effective sample sizes exceeding 200 for all parameters across all runs. Based on the consistency of results across runs and comparative summary statistics, we selected the fourth run for our final divergence time estimates (Supplementary Table 22).

## Reporting summary

Further information on research design is available in the Nature Portfolio Reporting Summary linked to this article.

## Data availability

All genome data analysed in this study are available from public databases, including UniProt, NCBI, Ensembl and other resources. The specific publications and download links for the 154 genomes are provided in Supplementary Table 1.

## Code availability

The computer code used in the analyses has been deposited in GitHub (https://github.com/JLWei7/animal_terrestrialisation).

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

**Acknowledgements** This work used ACRC HPC University of Bristol. We thank P. Holland, T. Williams and J. Vinther for their comments and suggestions on the analyses. J.W. is supported by China Scholarship Council-University of Bristol joint-funded Scholarship (202206350023). M.Á.-P. and J.P. are supported by the Wellcome Trust (210101/Z/18/Z) and the School of Biological Sciences (University of Bristol). M.Á.-P. was also supported by a fellowship from the Fundación General CSIC´s ComFuturo, which received funding from the European Union's Horizon 2020 research and innovation programme under the Marie Skłodowska-Curie grant agreement 101034263. M.Á.-P. is recipient of Ramon y Cajal grant RYC2023-043807-I of Spanish Ministry of Science, Innovation and Universities (MCIN/AEI/ and El FSE). P.C.J.D. is supported by Gordon and Betty Moore Foundation grant GBMF9741, Leverhulme Trust Research Fellowship grant RF-2022-167, Biotechnology and Biological Sciences Research Council grants BB/T012773/1 and BB/Y003624/1. D.P. is supported by Gordon and Betty Moore Foundation grant GBMF9741 and Leverhulme Research Project Grant RPG-2024-030.

**Author contributions** Conceptualization: J.W., D.P., P.C.J.D., M.Á.-P. and J.P. Methodology: J.W., M.Á.-P. and J.P. Data analysis: J.W. Visualization: J.W. Writing, original draft: J.W., M.Á.-P. and J.P. Writing, review and editing: J.W., D.P., P.C.J.D., M.Á.P. and J.P. Supervision: D.P., P.C.J.D., M.Á.-P. and J.P.

**Competing interests** The authors declare no competing interests.

**Additional information**

**Correspondence and requests for materials** should be addressed to Jordi Paps.

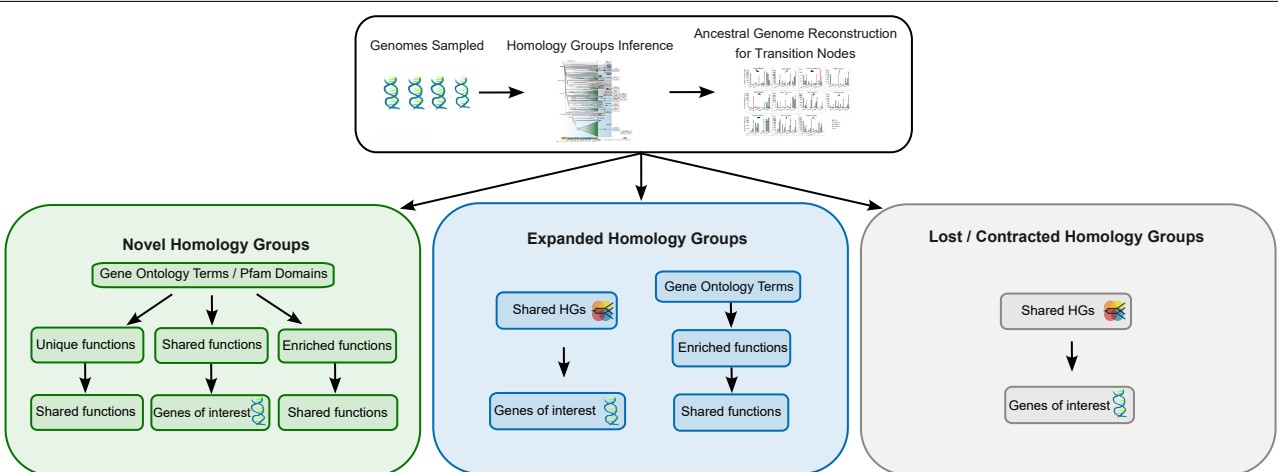

**Extended Data Fig. 1 | Overview of the InterEvo (Intersection Framework for Convergent Evolution) workflow.** The pipeline used in this study comprises three main analyses following homology group (HG) inference and ancestral genome reconstruction. Analyses of expanded HGs, contracted HGs, and lost HGs involve identifying HGs shared across transition nodes. Analyses of novel and expanded HGs involve identifying shared biological functions (including Gene Ontology (GO) terms and Pfam domains) across transition nodes, followed by enrichment analysis. The workflow integrates comparative genomics, homology inference, and functional annotation to detect convergent evolution patterns across independent terrestrialisation events.

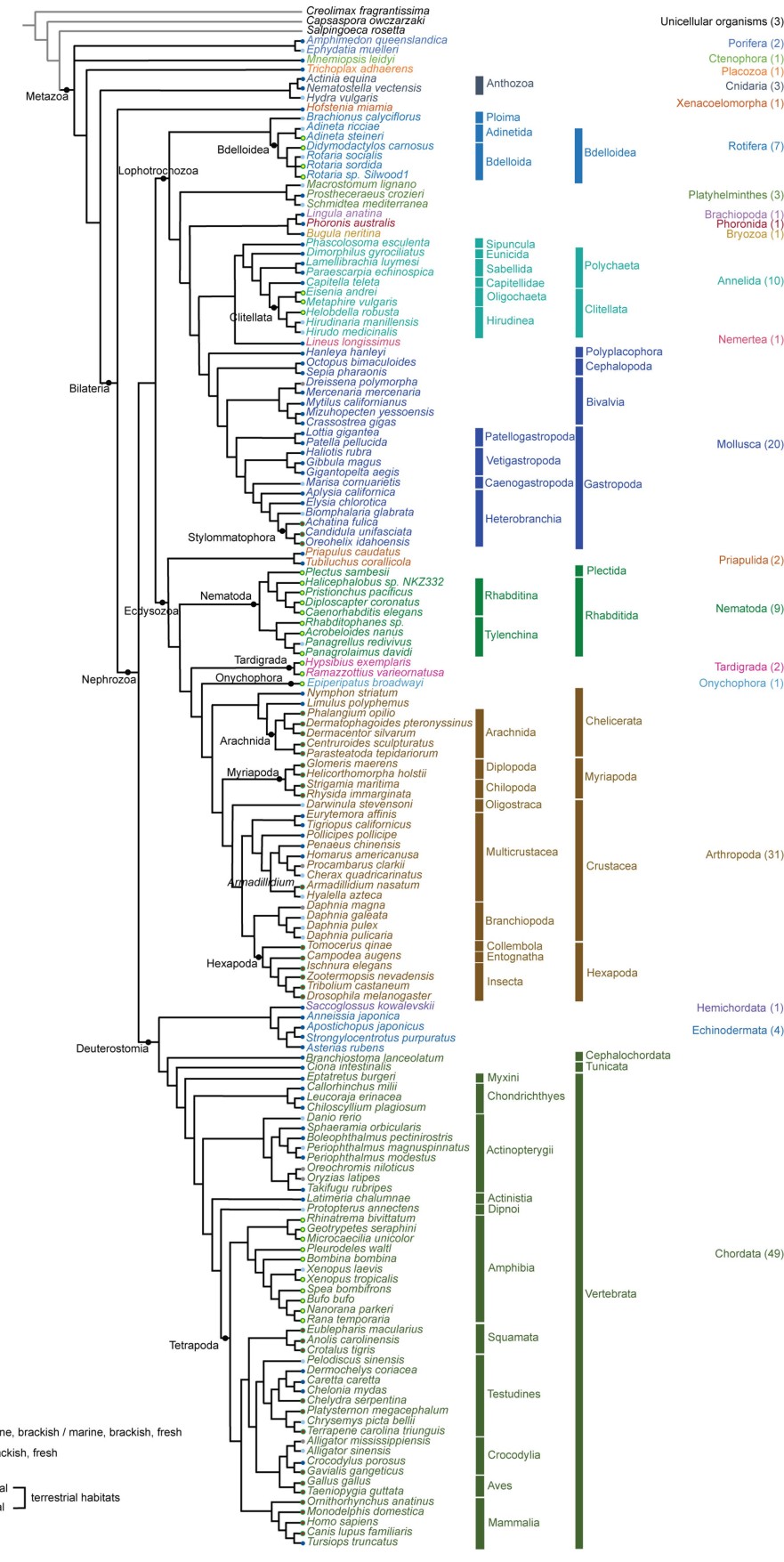

**Extended Data Fig. 2 | Species tree of the 154 sampled taxa.** Species tree of the 154 taxa sampled in this study. The habitat types are indicated as follows: marine (dark blue nodes), brackish (grey nodes), freshwater (light blue nodes), semi-terrestrial (yellow circles with green outlines), and fully terrestrial (red circles with green outlines).

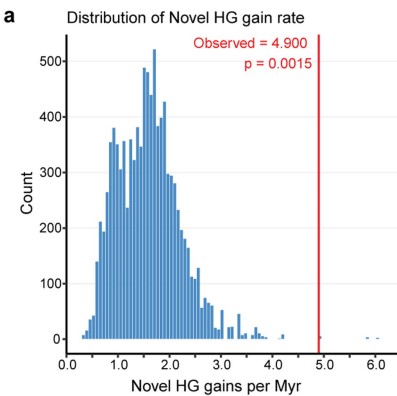
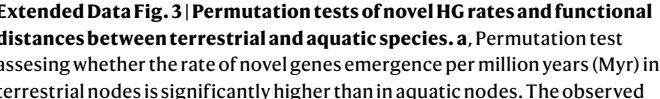
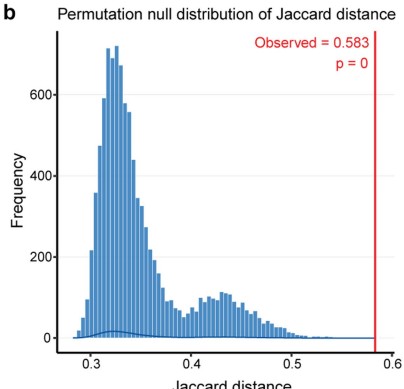

**Extended Data Fig. 3 | Permutation tests of novel HG rates and functional distances between terrestrial and aquatic species. a**, Permutation test assesing whether the rate of novel genes emergence per million years (Myr) in terrestrial nodes is significantly higher than in aquatic nodes. The observed terrestrial rate is indicated by the red bar. **b**, Permutation test assesing whether the biological functions in terrestrial nodes are significantly different from those in other nodes The observed GO distance between terrestrial and aquatic groups is indicated by the red bar. See Supplementary Text 1.1 for details.

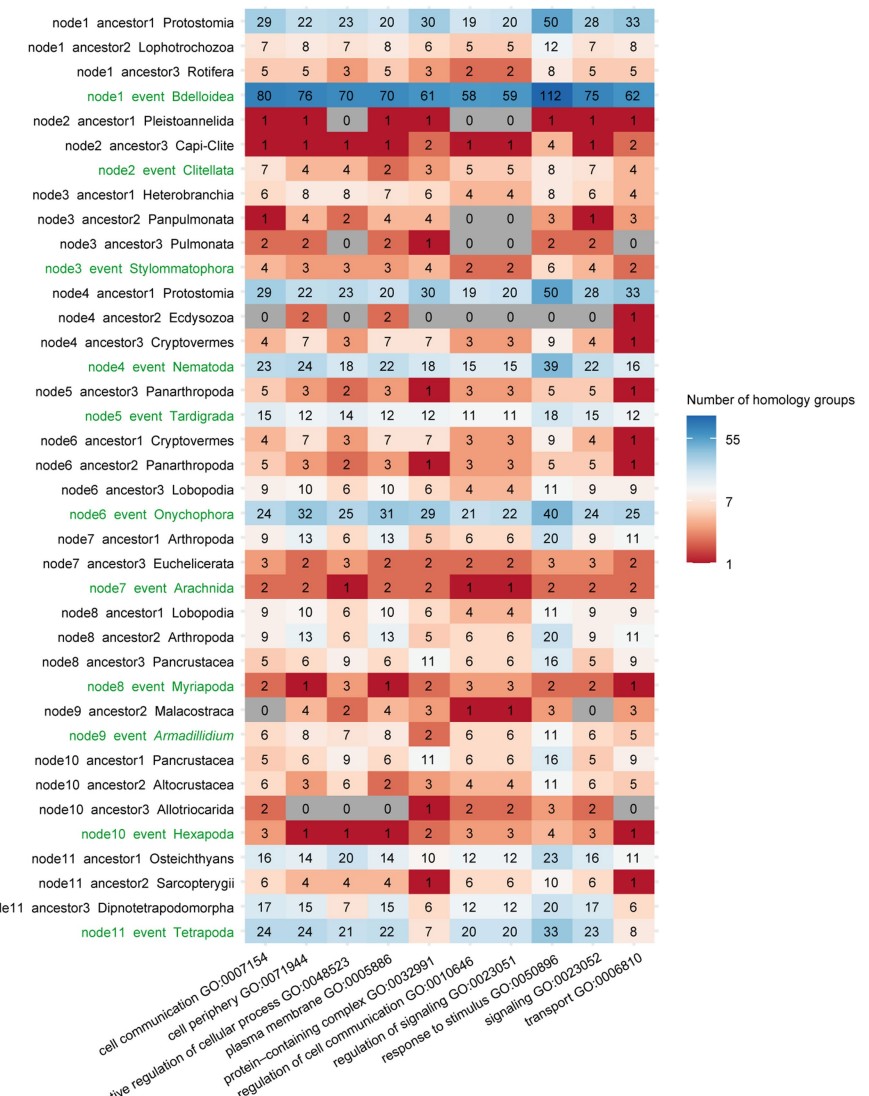

**Extended Data Fig. 4 | Novel homology group counts across key functions in terrestrial nodes and their ancestors.** The heatmap compares the number of novel HGs associated with 10 most specific GO terms in terrestrial nodes and their three immediate ancestors. These 10 GO terms were selected from the bottom-level hierarchy of the 27 GO terms of novel genes shared across all terrestrial nodes. Terrestrial nodes are highlighted in green text, and their ancestor nodes are shown in black. Columns represent GO terms, and cells show the number of novel HGs associated with each term. The colour gradient from red (low) through white to blue (high) represents the log-transformed values of HG numbers to improve visualisation of differences in scale.

**Extended Data Table 1 | Representative genes associated with terrestrialisation-linked GOs in human and fruit fly**

| | Novel Genes associated with terrestrialisation-linked GOs in human | | |
|---|---|---|---|
| Gene Symbol | Protein Name | Protein Class | Biological Functions |
| APOA2 | Apolipoprotein A-II | transfer/carrier protein (PC00219) | lipid metabolism |
| IL27 | Interleukin-27 subunit alpha | | |
| OSM | Oncostatin-M | | |
| XCL1 | Lymphotactin | | |
| XCL2 | Cytokine SCM-1 beta | intercellular signal molecule (PC00207) | |
| CXCL16 | C-X-C motif chemokine 16 | | |
| TNFSF18 | Tumor necrosis factor ligand superfamily member 18 | | |
| FLT3LG | Fms-related tyrosine kinase 3 ligand | | immunity and response to stimuli |
| CD1A, CD1B, CD1C, CD1E | T-cell surface glycoprotein | defense/immunity protein (PC00090) | |
| CD1D | Antigen-presenting glycoprotein CD1d | | |
| TMIGD2 | Transmembrane and immunoglobulin domain-containing protein 2 | cell adhesion molecule (PC00069) | |
| PLAUR | Urokinase plasminogen activator surface receptor | transmembrane signal receptor (PC00197) | |
| LYPD3 | Ly6_PLAUR domain-containing protein 3 | | |
| MPIG6B | Megakaryocyte and platelet inhibitory receptor G6b | | blood cell function regulation |
| SPP1 | Osteopontin | intercellular signal molecule (PC00207) | bone regeneration |
| ENAM | Enamelin | structural protein (PC00211) | teeth development |
| GPR152 | Probable G-protein coupled receptor 152 | transmembrane signal receptor (PC00197) | retinal cell-to-cell communication |
| AKAP3, AKAP4, AKAP5 | A-kinase anchor protein | scaffold/adaptor protein (PC00226) | reproductive strategies |
| DKKL1 | Dickkopf-like protein 1 | membrane traffic protein (PC00150) | |
| ZNF239 | Zinc finger protein 239 | gene-specific transcriptional regulator (PC00264) | |
| TBC1D21 | TBC1 domain family member 21 | protein-binding activity modulator (PC00095) | |
| PPP1R3F | Protein phosphatase 1 regulatory subunit 3F | | neurodevelopment |
| HR | Lysine-specific demethylase hairless | chromatin/chromatin-binding, or -regulatory protein (PC00077) | hair-cycle regulation (suggesting skin barrier) |
| | Novel Genes associated with terrestrialisation-linked GOs in fruit fly | | |
| Pof | Protein painting of fourth | RNA metabolism protein (PC00031) | reproductive strategies |
| MESR4 | Misexpression suppressor of ras 4, isoform A | gene-specific transcriptional regulator (PC00264) | |
| Ir64a, Ir75d, Ir31a, Ir84a | Ionotropic receptor | transmembrane signal receptor (PC00197) | sensory activity (response to stimuli) |
| Gr39b | Putative gustatory receptor 39b | | |

The table lists gene symbols, protein names, protein class (annotated using PANTHER 19.0[80]) and their functions corresponding to the 55 "most specific" GO terms identified in the terrestrial adaptation analysis, with novel genes in human and fruit fly provided as examples.

# Reporting Summary

## Statistics

For all statistical analyses, confirm that the following items are present in the figure legend, table legend, main text, or Methods section.

| n/a | Confirmed | |
|---|---|---|
| ☐ | ☒ | The exact sample size (*n*) for each experimental group/condition, given as a discrete number and unit of measurement |
| ☐ | ☒ | A statement on whether measurements were taken from distinct samples or whether the same sample was measured repeatedly |
| ☐ | ☒ | The statistical test(s) used AND whether they are one- or two-sided<br>*Only common tests should be described solely by name; describe more complex techniques in the Methods section.* |
| ☐ | ☒ | A description of all covariates tested |
| ☐ | ☒ | A description of any assumptions or corrections, such as tests of normality and adjustment for multiple comparisons |
| ☐ | ☒ | A full description of the statistical parameters including central tendency (e.g. means) or other basic estimates (e.g. regression coefficient) AND variation (e.g. standard deviation) or associated estimates of uncertainty (e.g. confidence intervals) |
| ☐ | ☒ | For null hypothesis testing, the test statistic (e.g. *F*, *t*, *r*) with confidence intervals, effect sizes, degrees of freedom and *P* value noted<br>*Give P values as exact values whenever suitable.* |
| ☐ | ☒ | For Bayesian analysis, information on the choice of priors and Markov chain Monte Carlo settings |
| ☒ | ☐ | For hierarchical and complex designs, identification of the appropriate level for tests and full reporting of outcomes |
| ☒ | ☐ | Estimates of effect sizes (e.g. Cohen's *d*, Pearson's *r*), indicating how they were calculated |

*Our web collection on statistics for biologists contains articles on many of the points above.*

## Software and code

Policy information about availability of computer code

| Data collection | Publicly available genomes are listed in the supplementary table 1. |
|---|---|
| Data analysis | Cd-hit v4.8.1<br>BUSCO v5.4.7<br>Orthofinder v2.5.5<br>MAFFT v7.505<br>DIAMOND v2.1.8<br>IQ-TREE v2.2.2.6<br>trimAl v1.4.rev15<br>FASconCAT-G v1.05.1<br>CAFE5<br>Phylogenetically Aware Parsing Script<br>PAML v4.10.7<br>All scripts available in GitHub https://github.com/JLWei7/animal_terrestrialisation |

For manuscripts utilizing custom algorithms or software that are central to the research but not yet described in published literature, software must be made available to editors and reviewers. We strongly encourage code deposition in a community repository (e.g. GitHub). See the Nature Portfolio guidelines for submitting code & software for further information.

## Data

Policy information about availability of data

All manuscripts must include a data availability statement. This statement should provide the following information, where applicable:

- Accession codes, unique identifiers, or web links for publicly available datasets
- A description of any restrictions on data availability
- For clinical datasets or third party data, please ensure that the statement adheres to our policy

Publicly available genomes are listed in the supplementary table 1.

## Research involving human participants, their data, or biological material

Policy information about studies with human participants or human data. See also policy information about sex, gender (identity/presentation), and sexual orientation and race, ethnicity and racism.

| | |
|---|---|
| Reporting on sex and gender | *Use the terms sex (biological attribute) and gender (shaped by social and cultural circumstances) carefully in order to avoid confusing both terms. Indicate if findings apply to only one sex or gender; describe whether sex and gender were considered in study design; whether sex and/or gender was determined based on self-reporting or assigned and methods used. Provide in the source data disaggregated sex and gender data, where this information has been collected, and if consent has been obtained for sharing of individual-level data; provide overall numbers in this Reporting Summary. Please state if this information has not been collected. Report sex- and gender-based analyses where performed, justify reasons for lack of sex- and gender-based analysis.* |
| Reporting on race, ethnicity, or other socially relevant groupings | *Please specify the socially constructed or socially relevant categorization variable(s) used in your manuscript and explain why they were used. Please note that such variables should not be used as proxies for other socially constructed/relevant variables (for example, race or ethnicity should not be used as a proxy for socioeconomic status). Provide clear definitions of the relevant terms used, how they were provided (by the participants/respondents, the researchers, or third parties), and the method(s) used to classify people into the different categories (e.g. self-report, census or administrative data, social media data, etc.) Please provide details about how you controlled for confounding variables in your analyses.* |
| Population characteristics | *Describe the covariate-relevant population characteristics of the human research participants (e.g. age, genotypic information, past and current diagnosis and treatment categories). If you filled out the behavioural & social sciences study design questions and have nothing to add here, write "See above."* |
| Recruitment | *Describe how participants were recruited. Outline any potential self-selection bias or other biases that may be present and how these are likely to impact results.* |
| Ethics oversight | *Identify the organization(s) that approved the study protocol.* |

Note that full information on the approval of the study protocol must also be provided in the manuscript.

# Field-specific reporting

Please select the one below that is the best fit for your research. If you are not sure, read the appropriate sections before making your selection.

☐ Life sciences　　　☐ Behavioural & social sciences　　　☒ Ecological, evolutionary & environmental sciences

For a reference copy of the document with all sections, see nature.com/documents/nr-reporting-summary-flat.pdf

# Ecological, evolutionary & environmental sciences study design

All studies must disclose on these points even when the disclosure is negative.

| | |
|---|---|
| Study description | We apply comparative genomics pipeline to explore the role of convergence and contingency in the evolutionary response of animal genomes to the process of terrestrialisation, and establish the timeline of the animal conquests of land. This study uncover that independent terrestrial events were driven by the emergence of similar biological functions. The timeline supports three temporal windows of land colonisation by animals during the last 487 My. While each lineage exhibits distinct adaptations, there is strong evidence of convergent genome evolution across the Animal Kingdom suggesting that, in large part, adaption to life on land is predictable, linking genes to ecosystems. |
| Research sample | This study conducted thorough comparative genomic analyses across 154 canonical genomes, of which 151 from 21 animal phyla and three from non-animal holozoans. |
| Sampling strategy | Sample size was collected as many as animal phyla. The taxa sampling represents the diversity of animals and focuses on species flanking nodes representing terrestialisation events. |

| Data collection | All available genomes were downloaded through Uniprot, EnsEMBL, NCBI or publicly available genome browsers.  Sources of genomes are listed in the supplementary table 1. |
|---|---|
| Timing and spatial scale | All genomes were downloaded in May 2023. |
| Data exclusions | No data was excluded from the analyses. |
| Reproducibility | The scripts and pipeline used by this study is available at Github (https://github.com/JLWei7/animal_terrestrialisation). All steps are reproducible. |
| Randomization | The organisms were chosen to ensure broad and balanced coverage across aquatic and terrestrial lineages from animals for comparative analysis. The topology was based on the tree position of species inferred by previous literature |
| Blinding | Blinding was not relevant to this study as it involved comparative genomic analyses. There was no observer bias might influence outcomes. |

Did the study involve field work?    ☐ Yes    ☒ No

# Reporting for specific materials, systems and methods

We require information from authors about some types of materials, experimental systems and methods used in many studies. Here, indicate whether each material, system or method listed is relevant to your study. If you are not sure if a list item applies to your research, read the appropriate section before selecting a response.

## Materials & experimental systems

| n/a | Involved in the study |
|---|---|
| ☒ ☐ | Antibodies |
| ☒ ☐ | Eukaryotic cell lines |
| ☒ ☐ | Palaeontology and archaeology |
| ☒ ☐ | Animals and other organisms |
| ☒ ☐ | Clinical data |
| ☒ ☐ | Dual use research of concern |
| ☒ ☐ | Plants |

## Methods

| n/a | Involved in the study |
|---|---|
| ☒ ☐ | ChIP-seq |
| ☒ ☐ | Flow cytometry |
| ☒ ☐ | MRI-based neuroimaging |

## Plants

| Seed stocks | *Report on the source of all seed stocks or other plant material used. If applicable, state the seed stock centre and catalogue number. If plant specimens were collected from the field, describe the collection location, date and sampling procedures.* |
|---|---|
| Novel plant genotypes | *Describe the methods by which all novel plant genotypes were produced. This includes those generated by transgenic approaches, gene editing, chemical/radiation-based mutagenesis and hybridization. For transgenic lines, describe the transformation method, the number of independent lines analyzed and the generation upon which experiments were performed. For gene-edited lines, describe the editor used, the endogenous sequence targeted for editing, the targeting guide RNA sequence (if applicable) and how the editor was applied.* |
| Authentication | *Describe any authentication procedures for each seed stock used or novel genotype generated. Describe any experiments used to assess the effect of a mutation and, where applicable, how potential secondary effects (e.g. second site T-DNA insertions, mosiacism, off-target gene editing) were examined.* |

