## [Peer Review File · Nature]

Convergent genome evolution shaped the emergence of terrestrial animals

Corresponding Author: Dr Jordi Paps

Version 1:

Reviewer comments:

Referee #1

(Remarks to the Author)

In "Convergent genome evolution shaped the emergence of terrestrial animals", Wei and co-authors investigate the genomic basis for animal terrestrialisation. The colonisation of land by eukaryotes is one of the most transformative events in the history of our planet. While much work has focused on how this event occurred in plants, much less is known about how animals transitioned into terrestrial environments. However, this has occurred repeatedly in different animal phyla, from tetrapods and arthropods to land snails, earthworms, and other less familiar taxa, such as water bears, velvet worms, and ribbon worms. To address this knowledge gap, the authors capitalise on the increase in the number of reference genomes to assemble an extensive dataset of 154 genomes, encompassing 20 different animal phyla and targeting 11 independent transitions into terrestrial environments. Combining gene family evolutionary analyses with molecular clock analyses, the authors conclude that these 11 terrestrialisation events occurred in parallel around three main geological periods (an old one mostly involving ecdysozoans, a mid one involving tetrapods and earthworms, and a more recent one involving rotifers and pulmonate snails) and cooccurred with distinct changes in gene family composition. While many of these changes appear lineage-specific, the authors also identify changes affecting similar biological functions (e.g., osmotic regulation, ion regulation, sensory detection), supporting the hypothesis that the selection pressures resulting from parallel transitions into land resulted in similar adaptations. Surprisingly, the authors identify the loss of regenerative potential and genes as one of these shared traits.

Altogether, the study represents a commendable effort to address a crucial outstanding question in animal evolution. The methods are sound, but there are important aspects that the authors need to consider to strengthen the conclusions, interpretations, broader biological impact and novelty of their data.

1) The authors use the power of 11 parallel transitions to define changes associated with terrestrialisation. How statistically significant are these changes? In other words, if randomly selecting 11 nodes in their phylogeny at equivalent phylogenetic distances, would they see something similar? How does gene content change within the terrestrial clade, and to what extent can some of the biological functions/genes that the authors consider critical for the transition to land be lost within already terrestrial lineages (i.e., the expansion/gain at the node could have been more contingent than adaptive)? Likewise, the sister lineages to these terrestrial clades are often found in freshwater environments. Are the changes in some of the biological functions found in terrestrial clades also associated with transitions to freshwater environments (e.g., osmotic and ion regulation)? Could these changes predate the transition to land? In short, the manuscript gives the impression that well-defined bursts of gene repertoire change underpin terrestrialisation, but it is unclear whether alternative scenarios are properly assessed (e.g., something more gradual, contingent and stochastic, where some changes already existed and were exapted, others that occurred at the node were subsequently lost, and the genetic fine tuning to a terrestrial environment happened later, as the groups became strictly terrestrial).

2) The conclusion that terrestrialisation affected regeneration is poorly supported (loss of a couple of gene families) and does not align with observations. Earthworms, and clitellates broadly, have amazing whole-body regenerative capacities, as land snails do and many tetrapods. The absence of good whole-body regeneration is general to ecdysozoans, likely due to the presence of an external cuticle that hampers quick epidermal closure and wound healing, rather than terrestrialisation.

Likewise, Bdelloid rotifers are extremely resistant, but eutelic, and their somatic cells do not undergo cell division after maturity, which probably impacts regeneration.

3) Following the point above, the general interpretations of how changes in biological functions, inferred through gene content variations at the nodes when terrestrialisation events occurred, might have driven morphological adaptations are feeble (e.g., lines 345-348). For example, Hexapods did not evolve moulting, vision or hard waterproof exoskeletons. These traits were already present in aquatic relatives and proved valuable, enabling multiple independent transitions to land (i.e., exaptations), even when they later became refined with the evolution of strictly terrestrial lineages. It is also unclear how clitellates adapted their nervous and muscular systems to terrestrial environments. The references provided do not address this and instead refer to meiobenthic organisms (subject to a whole other set of morphological adaptations), some of which have still unclear phylogenetic placements. As the authors acknowledge, reality is probably more nuanced, and transitions to land were not likely radical events that occurred at a single node, but rather gradual events resulting from lineages adapting to intertidal, brackish, and freshwater environments. Some of this could be explored using the assembled dataset, as indicated in point 1.

4) The authors use Gene Ontology annotation based on gene homology as the criterion to infer biological function. However, this approach is likely to be affected by the poor annotation of non-model systems and a lack of understanding of the exact functions of many genes in these systems, as acknowledged in the manuscript. This is likely a bigger issue with terminal differentiation genes, where the physiological roles of certain genes in model systems, from which the GO annotations are derived, may not have a clear equivalent in other systems. While there is not much the authors can do about this, I wonder if a GO annotation based on protein domains instead of gene orthology would have been more accurate or would change the conclusions of the study (i.e., explore the protein domains enriched/contracted/lost in those nodes and the biological functions associated to those protein domains).

6) Line 237: "Most share adaptations between fully terrestrial lineages are found in Arthropoda". This is expected, as it involves parallel transitions in more closely related taxa, suggesting that contingency (i.e., tinkering with what existed before) played a significant role in the transitions to land. In general, what is the impact of phylogenetic relatedness on the conclusions of the work?

7) Line 304–309: The colonisation of land by annelids and tetrapods is convergent (i.e. they occurred independently and implied a sea-to-land change). However, the authors suggest that it also involved convergent or similar genetic changes, based on the shared expansion of two gene families: disintegrin and ADAM proteins. Is this sufficient to deem convergence? Also, it is unclear how these two changes would relate to the biological adaptations mentioned in the text (more efficient fertilisation and complex nervous systems). The clitellate nervous system is not more complex than those of marine relatives (e.g. capitellids), and while clitellates evolved more complex fertilisation mechanisms (e.g. clitelium), the early tetrapods (amphibians) retained external fertilisation, and internal fertilisation only evolved later with a strict terrestrialisation. The authors try to establish strong connections between genetic changes that occurred at a node defined by a specific taxonomic sampling to explain a complex adaptive process that likely resulted from many gradual changes before and after that node. While some of this is understandable given the available data, it also presents an unnecessarily biased view towards one evolutionary scenario, especially when the dataset the authors assemble could allow them to explore different hypotheses, as discussed in some of the points above.

Other minor points:

- Figure 1: Phoronozoa or Brachiozoa?
- Line 25: Kingdom suggesting (space missing)
- Line 34: Terrestrial clades also have many developmental adaptations (e.g., encapsulated larvae, brooding, etc.)
- Line 37: genotype (typo)
- Line 67: were available
- Line 128: ENAM should go in italics if referring to the gene.
- Line 330: ". We found..." (typo)
- Line 384: "events more precisely or language models..." [remove stop after precisely]
- Line 412: "we selected some genomes THAT do not perfectly..."
- Line 418: Homo sapiens in italics.

(Remarks on code availability)

The repository includes all expected files, explanations and basic code.

Referee #2

(Remarks to the Author)

My comments are organised according to the list asked for on the review website, though combined in some areas to avoid repetition

A. Summary of the key results

B. Originality and significance: if not novel, please include reference

In this paper the authors approach an important question in the evolutionary history of life on earth, when and the genetic basis of how animals adapted to life on land. We already know that this has happened not so many times, with the successful lineages radiating to produce, alongside the other groups of life, (microbes, plants fungi) the terrestrial ecologies

we see today. There have been many studies linking specific genes to specific adaptive traits and asking if these evolve in parallel. However these are generally relatively simple traits wing colour patterns, vertebrate limb reduction/loss, and similar. Terrestrialisation is a different level of adaptation, involving many changes in anatomy, sensory-neural systems, biochemistry and physiology, and their integration. To my knowledge, while there have been studies looking at terrestrialisation in individual lineages, this is the first to take a broad comparative approach across the animal tree and to do so in a systematic way. I think this makes it an important general topic of potential wide interest.

The authors use a pipeline of their making (and previously used and published by them in studies identifying genetic changes at other key evolutionary nodes) to compare the proteomes of animals to identify gene content changes mapping onto nodes where animals shifted to land. The approach is to define 'homology groups': it has caveats but so do other approaches of correlating genes to adaptations at deep evolutionary nodes, and I think it's the one least affected by caveats. Importantly in my opinion, the approach is both systematic and quantitative, allowing Gene Ontology to be used to compare lineages/nodes. They also use the sequence data to build a time calibrated tree of the lineages involved. Following this is mapping numbers of gains and losses onto the nodes of the tree, including applying a method to determine when there have been expansions or contractions of gene number within homology groups, and whether there are shared patterns to this.

Defining these numbers and the gene groups that comprise them is an important step although I have a concern about the methodology I detail below. The main outcomes beyond this are proposing three episodes when territorialisation happened, and identification of a range of candidate genes that may reflect functions involved in the adaptation under study. As detailed below, I did find the first of these questionable, and the second, while individual cases are believable, is potentially subject to my concern above and has the appearance of being a selectively picked list of intriguing but untested associations.

In summary this is a potentially exciting topic but I do have some significant concerns around methodology, interpretation and the conclusions drawn

- C. Data & methodology: validity of approach, quality of data, quality of presentation
- D. Appropriate use of statistics and treatment of uncertainties
- E. Conclusions: robustness, validity, reliability

Here I deal with my major comments on the manuscript

Line 88 and onwards. I have a methodological concern over the data presented here, how they are interpreted and how this feeds into subsequent figures. As I understand the analysis and data summarized in Figure 1b, the numbers show the gene (homology group) changes that map onto these respective nodes. That's fine, and important to have a consistently applied methodological approach to identifying these. However surely these numbers will also be dependent on the gap between the base of the node and the next outgroup branch? I apologise if the authors have explicitly addressed this somewhere in the methods and I missed it, but I think we have to assume that there will be a baseline degree of gene turnover, and hence that a long unbroken branch will generally have higher numbers than a shorter one. For example if we then look at the data in figure 1b, this seems to fit with woodlice and bdelloid rotifers having high numbers, correlating with the time tree above that shows relatively recent crowns predated by long unbroken branches. And vice versa for the tetrapod and the other arthropod crowns. I understand that there are inherent limitations in such analyses due to what genomes are available and fundamentally to extinction of lineages that would break a branch, but I am struggling to see how we can then read much into the shared changes identified in Figure 3.

Figure 3a shows shared gene groups between the various combinations of terrestrialised lineages. It took a bit of time to work out from the text that the authors had used CAFE5 to build the lists of expansion/contraction at each node, this wasn't mentioned till the discussion. Understanding this after some effort, I do appreciate that this software is more sophisticated than just counting, and does model rates in some form. But, if we have nodes that have accumulated lots of change in part due to long unbroken branches preceding them, are we not more likely to see overlap in expansion/contraction of gene groups as a consequence of the higher numbers? See also my comment regarding line 152 below. I am not arguing that changes linked to terrestrialisation won't be captured here as well, I am pretty sure they will, but I am concerned the quantitative approach taken by the authors in this study, which is admirable as I say elsewhere, is compromised by not disentangling these. Does this not then feed in to undermining the GO distribution analyses as well?

Line 152: this relates to my comment above. The gene types identified here make sense in terms of adaptation, but I think some of them are well known as families that undergo relatively rapid changes in copy number in many lineages, most unrelated to terrestrialisation. For example the variable numbers of glutathione s-transferases in insects and cephalopods, of olfactory receptors in many lineages. How can we conclude the cross-lineage similarities observed here aren't a consequence of this general high lability and not the specific adaption under study here? If they are genes likely to change in copy number anyway, wont they be more likely to show as shared between nodes that have long preceding branches?

Line 200: here the authors divide the lineages into semi and fully terrestrial. Im in two minds about this. There are obviously differences in this 'axis' of life history, and quantifying it isn't possible so qualitative subdivision is justifiable. But might another axis/subdivision make more biological sense, eg between miniature interstitial animals and others? Ones with big brains/sense organs and others? Or does this confuse terrestrialisation with adaptation to micro versus macro life? I don't have a strong view, but do think the authors need to explain why the subdivision they have chosen is the right one.

Line 274 onwards. This section of the results concerns the relative timing of the terrestrialisation events. I have no concern about the time tree analysis shown in figure 1, the methods look robust. I am less persuaded about the definition of three episodes. Inevitably for this sort of analysis the probability windows for when these changes happened are quite large.

Overlapping terrestrialisations could be fully congruent. Or they could be separated by many millions of years. This means they could be emerging in similar ecological and climatic environments, or not. How then are we to interpret a small number of shared gene changes, like for example the ADAM protease expansions in tetrapods and clitellates, when we also think about their separation into semi versus fully terrestrial earlier on in the paper, and their adaptation into (I think, im not an expert in the paleo evidence here) burrowing versus mobile predatory life histories? I am struggling to see how we move from an interesting observation to deeper explanatory understanding especially given the comments I made above. I don't think the manuscript makes this leap.

F. Suggested improvements: experiments, data for possible revision
These are in addition to the above and really reflect minor comments

Abstract

In lines 14-15 there is a claim made that this approach allows one to decipher the roles of contingency and convergence in adaptation, however I don't think this is actually discussed in the manuscript
Late on, the abstract mentions regeneration. This is one small thing mentioned at one point in the paper and as detailed below I am sceptical as to its significance anyway, I don't think it should be in the abstract.

Introduction

Line 29: a small thing, but calling it 'one of the most iconic episodes' is counter to previous views and this paper's conclusions that territorialisation happened at multiple times spread over the last ~500 million years.
Line 33: similarly, I don't think we can claim these lineages faced the same ecological challenges. Animals adapting to terrestrial live at different times faced hugely different ecologies dependent on what had already territorialised and diversified previously.
Line 47: Delete vast here. As well as unnecessary, in this age 147 genomes isn't vast!

Line 107: this paragraph starts by stating something that hasn't been shown yet by the study. Its not wrong because they come on to mention this but it left me floundering to follow the reasoning for a bit till I read on a while.

Line 127: The use here of a couple of gene families subsequently then lost in a lineage returning to the water wasn't convincing to me. Seems cherry picked, there must be losses all over the tree on other nodes, why just pick on these two? Suggest either bolster this with a quantitative analysis or remove it.

Line 182: as for my comment on gene loss in cetaceans, two GO terms related to regeneration is not compelling. I could maybe be persuaded if there was good evidence that this homology group was involved in regeneration across multiple lineages, but as the authors discuss GO annotation is heavily skewed to a couple of lineages, and the genes themselves look like general tissue integrity and cellular system genes that have picked up regeneration GO annotation somewhere. I have a parallel thought on the chlorophyllase point below: I thought that fossil evidence was pretty clear that the Devonian aquatic ancestors of tetrapods were predators not herbivores? Happy to be corrected about that though.

Line 291: do the authors mean land environments here? Or is a landing environment something else, where they first came onto land maybe?

G. References: appropriate credit to previous work?
Yes I think so

H. Clarity and context: lucidity of abstract/summary, appropriateness of abstract, introduction and conclusions
I think these need a lot of work. The abstract had some really minor things given prominence and didn't emphasise the bigger aspects enough. The Intro is OK generally, but the discussion is long and disorganised. Mentioning the caveats with things like GO analysis is important, but I lost track of what the authors thought were their key outcomes in it.

(Remarks on code availability)

Referee #3

(Remarks to the Author)

This manuscript employs 154 complete genomes representative of the breadth of animal phylogeny in an analysis that concludes "there is strong evidence of convergent genome evolution..." with respect to territorialization events over the course of animal evolution. While the manuscript is attempting to address a broadly interesting evolutionary question, several aspects about this work render it problematic. Without digressing into excessive detail, some of the major concerns are mentioned here. However, before mentioning concerns about assumptions, analysis, or conclusions, the overall grammatical quality of the paper should be noted. This paper clearly was not well vetted before submission given the numerous glaring errors and typos.

The main concern is that the analyses undertaken in this work is not up to current molecular evolutionary or phylogenetic standards. Basically, the work builds a phylogenetic topology from the data at hand, uses a BLAST approach to identify genes or homology groups, and used GO terms for functional analyses. More discerning methods would include some type of phylogenetic independent contrasts approach augmented with individual gene trees for the HG to assess gene family (or HG) expansion and contraction. Just employing GO terms over such a broad swath of the animal kingdom is problematic as most of the function is drawn from a few select models (as noted on line 364). More attention to which findings were significant would have been appreciated (example line 150).

Other issues include:

- 1) There is no indication given of node support on the phylogeny. This is an issue as some branches vary from the current best estimates of animal phylogeny (e.g., byozoan placement, sponge placement [this is no longer controversial]). It is ok that these placements are not recovered, but the presentation needs to be clear on this issue and present analyses that will allow readers to assess the tree.
- 2) Lost HG are defined as "HG's lost in the all species with a node, while present in the sister groups or other species in outgroup." First there is only one sister group to a give node (assuming the node is resolved). Second this definition also defines HG's that are autapomorphies in the sister lineage or an outgroup lineage. That is, it contains more than "lost HG's" and autapomorphic HG's in sister taxa likely skew the calculations if done as presented.
- 3) In some instances of text appear contradictory. For example, the Abstract states, "there is strong evidence of genome evolution" but in the Introduction states, "...our results reveal that independent terrestrial events were driven by the emergence of similar biological function, although semi- and fully terrestrial lineages exhibit different patterns of genomic adaptation...". There are several other instances where the message is unclear or contradictory. One more is on line 89 which stated comparisons used immediate ancestors but elsewhere is it clear that numbers were compared between sister groups.
- 4) Better justification is needed as to why some clades were a focus (e.g. rotifers) and others were not (molluscs).
- 5) It is not clear if the gene families focused on were considered ahead of time or were post hoc. If the latter, there should be some discussion of how many gene families showed change. With enough comparisons some HG's will show significant differences. It is noted that a correction for multiple analyses was included in the GO term analyses.
- 6) There is a good bit of jargon and limited discussion in some cases to back up ideas. As a result, good chunks of the discussion seem ad hoc. For example lines 126-129 describing APOA2 and ENAM being involved in secondary transitions. Why these genes? This part of discussion seems a bit rambling. A better pretext discussing the genes and gene functions that would be needed, or lost, for terrestrialization should be given and then the work should specifically test for those genes or HG's.

(Remarks on code availability)

Version 2:

Reviewer comments:

Referee #1

(Remarks to the Author)

In this revised version of the manuscript entitled "Convergent genome evolution shaped the emergence of terrestrial animals", Wei and co-authors convincingly address the previously raised points. The statistical analyses of the changes in gene family composition are smart and strengthen the original observations. Likewise, the inclusion of gene functional enrichments based on PFAM annotation supports the original gene-based approaches, and the text is now generally more balanced and accurate. Overall, this is a fascinating study that addresses one of the crucial outstanding questions in animal evolution and will open new avenues for investigating animal adaptation to land at multiple levels, from palaeobiology to genetics, development, and cell biology.

I do not have further experimental concerns, but there are two main points the author should consider.

- I still find the connection between terrestrialisation and loss of regeneration poorly supported. While it might be true that terrestrial lineages exhibit, at least ancestrally, poor regeneration capacities, I do not see how the convergent loss of two gene families (dbl and plekstrin-related) can be proof of a genetic basis of that regeneration reduction. These are broad gene families that are involved in many biological processes, and the references provided to support their role in regeneration are limited to vertebrates and cell type regeneration, rather than organ/whole body regeneration. The interpretation as a "shift away from a more generalised aquatic larval regeneration" (line 437) is also unclear and based on a manuscript (REF 49) that presents a very unique hypothesis on the nature of regeneration (which is an ongoing debate in the field). Indeed, there is no such thing as "aquatic larval regeneration" as many aquatic systems with strong regenerative capacities do not have a larval stage (e.g., acoels). It is thus widely speculative that the loss of these two families is causally related to poor regenerative abilities across distantly phylogenetically related animal phyla, especially when the terrestrial lineages lacking those gene families likely share many other traits that could also be linked to the potential biological roles of these genes. Nevertheless, if the authors want to keep this interpretation, I suggest to remove it from the abstract (and use those few words to highlight other better supported points of the manuscript, such as the connection of the three windows of terrestrialisation with different geological/ecological conditions) and tone down/qualify the section about loss of regeneration (and probably remove the associated discussion), indicating that the function of these gene families in most terrestrial lineages is unknown and that their loss might be either a striking coincidence (ie., non-adaptive and by drift) or linked to other traits those lineages might share. To me, the convergent loss of chlorophyllase in terrestrial lineages seems more interesting as it might be related to a shift from an algal-rich diet to one based on land plants (again, speculative, but perhaps there is something in there that makes more sense than a vague connection of two broad gene families to neuronal/muscle cell regeneration in vertebrates).

- The text still needs some polishing and grammar correction, both in the main manuscript and in the supplementary

technical notes (there are many examples, but to pick a few: all figures are called as Figure. [with a dot at the end]; In line 111, the paragraph starts as "To infer this functional convergence", but that is a legacy of the previous version because the convergence has not been mentioned yet. Additionally, the text mixes British and American spelling, which should also be avoided. The Discussion is redundant at times and could be streamlined into two or three paragraphs that reflect on the key findings and limitations of the study.

Minor points:

- Lines 477–480: How does this study support the phylogenetic position of Xiphosura and Arachnida? There is no phylogenetic reconstruction in the study, and the phylogenetic position of any group should not be based on patterns of gene gain/loss.
- Finally, the GitHub repository might need to be updated with the code for the new analyses included in the manuscript.

Chema Martin

(Remarks on code availability)

The repository might need update with the latest analyses and scripts, but it is generally comprehensive and should ensure reproducibility.

Referee #2

(Remarks to the Author)

The authors have addressed the important methodological concerns I raised, especially regarding controlling for rates of gene turnover. I still find the emphasis on regeneration suspect, especially its inclusion in the abstract, when this is based on such a small number of genes with broad cellular functions. I think this could well be due to bias in how GO terms are attributed to genes. However, its not wrong for the authors to examine this and discuss, its just the emphasis and I would prefer it to be cut from the abstract.

(Remarks on code availability)

Responses to Referee #1

In "Convergent genome evolution shaped the emergence of terrestrial animals", Wei and co-authors investigate the genomic basis for animal terrestrialisation. The colonisation of land by eukaryotes is one of the most transformative events in the history of our planet. While much work has focused on how this event occurred in plants, much less is known about how animals transitioned into terrestrial environments. However, this has occurred repeatedly in different animal phyla, from tetrapods and arthropods to land snails, earthworms, and other less familiar taxa, such as water bears, velvet worms, and ribbon worms. To address this knowledge gap, the authors capitalise on the increase in the number of reference genomes to assemble an extensive dataset of 154 genomes, encompassing 20 different animal phyla and targeting 11 independent transitions into terrestrial environments. Combining gene family evolutionary analyses with molecular clock analyses, the authors conclude that these 11 terrestrialisation events occurred in parallel around three main geological periods (an old one mostly involving ecdysozoans, a mid one involving tetrapods and earthworms, and a more recent one involving rotifers and pulmonate snails) and cooccurred with distinct changes in gene family composition. While many of these changes appear lineage-specific, the authors also identify changes affecting similar biological functions (e.g., osmotic regulation, ion regulation, sensory detection), supporting the hypothesis that the selection pressures resulting from parallel transitions into land resulted in similar adaptations. Surprisingly, the authors identify the loss of regenerative potential and genes as one of these shared traits.

Altogether, the study represents a commendable effort to address a crucial outstanding question in animal evolution. The methods are sound, but there are important aspects that the authors need to consider to strengthen the conclusions, interpretations, broader biological impact and novelty of their data.

We thank Referee #1 for their positive assessment and for their insightful comments that have helped us improve the rigor and impact of our study. We hope they find the additional analyses and text changes following their advice make the findings stronger.

- **Point 1a:** *The authors use the power of 11 parallel transitions to define changes associated with terrestrialisation. **How statistically significant are these changes?** In other words, **if randomly selecting 11 nodes in their phylogeny at equivalent phylogenetic distances, would they see something similar?***

This is an excellent point regarding the statistical validation of our convergence analysis. Our observed data, which include terrestrial nodes nested within aquatic ones, lacks the statistical power for standard parametric tests (e.g., t-tests, ANOVA). Instead, we used phylogeny-wide **permutation tests**, similar to a bootstrap approach. Permutation tests determine the statistical significance of observed data by comparing it against an empirical null distribution generated from numerous random permutations of the original data (e.g., evolutionary rates of novel genes in a node, GOs in a node) with their labels (aquatic or terrestrial) reshuffled in each permutation. This allows us to test specific hypotheses about evolutionary patterns by breaking the observed correlation between variables while preserving their individual distributions.

In the **first permutation test**, we evaluated **if the rate of emergence of novel genes (number of novel genes emerging per million year) in terrestrial nodes was significantly higher than aquatic nodes**. We accounted for divergence times to address comments by reviewer #2, which can be found in the answers to reviewer #2 (please, see below). We collected the rate of emergence of novel genes for 11 terrestrial nodes and randomly selected 11 aquatic nodes (Actinopterygii, Ambulacraria, Bivalvia, Branchiopoda, Chondrichthyes, Cnidaria, Decapoda, Platyhelminthes, Priapulida, Sabellida and Vetigastropoda). We calculated the observed total evolutionary rate in the 11 terrestrial nodes as the total number of novel HGs divided by total divergence time (rate=4.900). We then repeatedly drew 11 sets of aquatic nodes randomly (with replacement) from the aquatic pool to build 10,000 permutations, and recomputing the evolutionary rate (total novel HGs counts divided by total divergence time), producing a null distribution of novel gene rates in aquatic nodes. As shown in Supplementary Figure 3a (also included here), the observed terrestrial rates (red bar) exceed the aquatic permutation rates (one-tailed $p = 0.0015$). Thus, the observed higher novel gene rates found in terrestrial lineages are significantly higher than in aquatic nodes even after correcting for divergence time. In contrast with novel HG inferred from our pipeline, the expanded and contracted genes were inferred with CAFE5, whose birth-death model is intrinsically scaled by branch length and do not require correction.

Supplementary Figure 3 Permutation tests of novel HGs counts and biological function distance. a) Permutation tests assess if the number of novel genes emerging per million years (Myr) in terrestrial nodes was significantly higher than aquatic nodes. b) Permutation tests assess if the biological functions found in terrestrial nodes are significantly different from those in other nodes.

The **second permutation test** assessed if the biological functions found in terrestrial nodes are significantly different from those in other nodes. We included lineages with the biggest taxon sampling from random aquatic lineages, including Actinopterygii, Ambulacraria, Bivalvia, Branchiopoda, Cnidaria, Decapoda and Platyhelminthes. First, we converted the Gene Ontology (GO) matrix derived from the novel genes for each lineage into a binary presence/absence matrix, then quantified the dissimilarity between terrestrial and aquatic GO term profiles by measuring the proportion of non-shared terms (Jaccard GO distance). Then we built 10,000 permutations, each one reshuffling the aquatic/terrestrial labels between lineages and recalculating the Jaccard distance between the reshuffled “aquatic” and “terrestrial lineages”. This generates a null distribution of GO distances between lineages whose habitat has been reshuffled (Supplementary Figure 3b, shown above). The GO distances for the real observed data (0.583, red bar, empirical $p < 1 \times 10^{-4}$) are outside the permutation distribution, indicating that around 42% of GO terms are habitat-specific and that this degree of gene turnover is highly unlikely under random node choice. This proves that GOs derived novel genes are functionally distinct between terrestrial and aquatic ones.

We thank the referee for suggesting these statistical analyses, as we believe they strengthen our results. Now we have added these analyses to our manuscript in line 101 and 144 and Supplementary Figure 3 and Supplementary Information 1.1

- **Point 1b: How does gene content change within the terrestrial clade, and to what extent can some of the biological functions/genes that the authors consider critical for the transition to land be lost within already terrestrial lineages (i.e., the expansion/gain at the node could have been more contingent than adaptive)?**

That is an intriguing question. Our study distinguishes between **novel HG** (gained in the last common ancestor of a lineage, but that can be lost later) and **novel core HGs** (novel genes that are consistently retained across a lineage, and thus we argue must be critical). Based on this referee’s question, we checked the difference between novel and novel core as it highlights the differences between critical and non-critical functions. We have now inferred the GOs for these terrestrial novel genes that are then lost, and added them to the manuscript and Supplementary data (manuscript line 145 and Supplementary Figure 9, Supplementary Table 6), please see below a couple of examples from Supplementary Figure 9a. For others, please see Supplementary Table 6.

Supplementary Figure 9. GO terms that are associated with novel genes but not with novel core genes, these functions were gained in the terrestrial ancestor but later lost. a) Examples of Hexapoda and Tetrapoda. b) Shared functions gained in the terrestrial ancestor but later lost. The blue bars are the GOs shared in four nodes.

We want to highlight that we identified shared critical functions retained in the members of a terrestrial lineage by mining the shared GOs between terrestrial novel core genes in the previous version of the manuscript (e.g., Figure 2 of the manuscript). We also have identified the shared GOs between novel terrestrial genes that are later lost (Supplementary Figure 9b).

- **Point 1c:** *Likewise, the sister lineages to these terrestrial clades are often found in freshwater environments. Are the changes in some of the biological functions found in terrestrial clades also associated with transitions to freshwater environments (e.g., osmotic and ion regulation)? Could these changes predate the transition to land?*

This is a valid point. We want to emphasise that the manuscript does not claim that all these biological functions are exclusively found in terrestrial nodes, and we agree some might be exaptations and shared with freshwater ancestors. For example, gene families expanded/contracted in terrestrial lineages (Figure 3) are shared with aquatic ancestors and relatives, so these pre-existing genes increased (or reduced) their gene repertoire, as this referee indicates. We have tried to clarify in the text in lines 85 and 460. For novel gene families, while their functions might be found among aquatic relatives or ancestors, their emergence is specific to terrestrial lineages; we believe the strength of our findings derives from the convergent evolution approach, showing that these functions repeatedly emerged or were co-opted independently in diverse terrestrial lineages. Finally, one of our analyses did investigate “Unique GO” and “Unique Pfams”, which are GO terms/Pfams associated with novel genes present in terrestrial nodes but absent in the GO terms/Pfams of their ancestor nodes (which include aquatic species). These can be found in line 135 of the manuscript and Supplementary Figure 6 and 7. We found Unique GO terms shared by at least five terrestrialisation events are related to metabolism and ion transport.

Supplementary Figure 7 Distribution of shared "Unique Pfams" across terrestrial nodes. The shared "Unique Pfams" in at least three nodes are labelled with green bars.

GO terms of novel genes shared between terrestrial and freshwater species

Biological Process

Cellular Component

Molecular Function

Supplementary Figure 8. GO terms of novel genes shared between terrestrial and freshwater species

We analysed the shared GO terms (Supplementary Figure 8) and shared PFAMs (Supplementary Table 5) of novel genes between terrestrial and freshwater species, mostly sister groups. We found freshwater functional terms overlapping with terrestrial ones, including GO terms such as response to stress, plasma membrane and protein-containing complex, and Pfams such as protein kinase and Trypsin (Supplementary Table 5), agreeing with the view that some major functional categories involved in terrestrialisation were present in freshwater ancestors and later refined for land adaptation.

We have now clarified our text to better reflect that many changes may represent exaptations and are found in freshwater lineages, rather than solely *de novo* gains in terrestrial nodes, we have added a sentence in line 143 to account for that.

- **Point 1d:** *In short, the manuscript gives the impression that well-defined bursts of gene repertoire change underpin terrestrialisation, but it is unclear whether alternative scenarios are properly assessed (e.g., something more gradual, contingent and stochastic, where some changes already existed and were exapted, others that occurred at the node were subsequently lost, and the genetic fine tuning to a terrestrial environment happened later, as the groups became strictly terrestrial).*

We appreciate the valid concerns of this referee and hope all the new analyses have addressed them.

- **Point 2:** *The conclusion that terrestrialisation affected regeneration is poorly supported (loss of a couple of gene families) and does not align with observations. Earthworms, and clitellates broadly, have amazing whole-body regenerative capacities, as land snails do and many tetrapods. The absence of good whole-body regeneration is general to ecdysozoans, likely due to the presence of an external cuticle that hampers quick epidermal closure and wound healing, rather than terrestrialisation. Likewise, Bdelloid rotifers are extremely resistant, but eutelic, and their somatic cells do not undergo cell division after maturity, which probably impacts regeneration.*

We thank the reviewer for this critical observation. The manuscript does not claim that regeneration has been totally lost in all the transitions to land, and it does not single out whole-body regeneration, but we argue that regeneration genes have been lost in many terrestrialization events. As indicated above, we believe convergent evolution is a great system to detect patterns, and we believe it is striking that the genes related to regeneration have been widely lost. Dbl gene family was lost in **8** out of 11 terrestrial events (retained only in bdelloids, Stylommatophora and tetrapods), and the pleckstrin-domain gene family lost in **7** out of 11 terrestrial events (retained only in bdelloids, Stylommatophora, myriapods and tetrapods; Supplementary Table 9). For reference, we don't find gene families convergently expanded or contracted in more than four nodes (Figure 3 in the main manuscript), so complete HG loss in 7 to 8 terrestrial nodes is remarkable. Our results therefore suggest a tendency, with repeated loss of regeneration components.

There might be exceptions (rotifers as mentioned by the reviewer) to this pattern, and some terrestrial subclades might have regained or retained regenerative capabilities (e.g. salamanders). We also concede the reviewer's point that the cuticle in ecdysozoans is a major constraint but argue that the transition to land imposes novel pressures on wound healing (e.g., microbial infection, desiccation) that would drive further, convergent evolution in these

pathways across all terrestrial groups. We would argue that in general, aquatic annelids (especially marine polychaetes) tend to have stronger and broader regenerative abilities than terrestrial annelids like earthworms and other clitellates, with polychaetes that can regenerate both anterior (head) and posterior (tail) segments while clitellates show reduced anterior regeneration¹⁻³(Ozpolat and Bely 2016; Bely et al 2014; Bely 2006).

We now explicitly discuss these cases in the main text lines 206–210 and 356–359. We propose that while some members of terrestrial lineages have remarkable regenerative abilities, the genomic changes at the ancestral terrestrial node may reflect a shift away from a more generalized aquatic larval regeneration program. The impressive abilities seen in modern forms are likely derived specializations.

- **Point 3:** *Following the point above, the general **interpretations of how changes in biological functions**, inferred through gene content variations at the nodes when terrestrialisation events occurred, **might have driven morphological adaptations** are feeble (e.g., lines 345-348). For example, Hexapods did not evolve moulting, vision or hard waterproof exoskeletons. **These traits were already present in aquatic relatives and proved valuable, enabling multiple independent transitions to land (i.e., exaptations)**, even when they later became refined with the evolution of strictly terrestrial lineages. It is also unclear how clitellates adapted their nervous and muscular systems to terrestrial environments. The references provided do not address this and instead refer to meiobenthic organisms (subject to a whole other set of morphological adaptations), some of which have still unclear phylogenetic placements. As the authors acknowledge, reality is probably more nuanced, and transitions to land were not likely radical events that occurred at a single node, but rather gradual events resulting from lineages adapting to intertidal, brackish, and freshwater environments. Some of this could be explored using the assembled dataset, as indicated in point 1.*

This is a very fair point, and we agree that the language could be clearer. As indicated above (Point 1c), we have revised the manuscript to more explicitly acknowledge the crucial role of **exaptation**. We agree that Hexapods did not evolve moulting, vision, or waterproof cuticles, but rather that their gene repertoire underlying these functions expanded, pointing to genomic changes driving this transition. We now clarify that traits like moulting and vision were present in aquatic ancestors, but that the gene family expansions/gains we identify at terrestrial nodes represent **innovations and refinements** of these pre-existing systems for life on land. For example, we discuss the expansion of specific gene families involved in the synthesis of the waxy layer of the exoskeleton, a key refinement for waterproofing in a terrestrial environment. We

have also removed the unsubstantiated claim about clitellate nervous systems and ensured all functional interpretations are more directly supported by the data or appropriate citations.

- **Point 4:** *The authors use **Gene Ontology annotation** based on gene homology as the criterion to infer biological function. However, this approach **is likely to be affected by the poor annotation of non-model systems** and a lack of understanding of the exact functions of many genes in these systems, as acknowledged in the manuscript. This is likely a bigger issue with terminal differentiation genes, where the physiological roles of certain genes in model systems, from which the GO annotations are derived, may not have a clear equivalent in other systems. While there is not much the authors can do about this, **I wonder if a GO annotation based on protein domains instead of gene orthology would have been more accurate** or would change the conclusions of the study (i.e., explore the protein domains enriched/contracted/lost in those nodes and the biological functions associated to those protein domains).*

This is an excellent suggestion. The program we used to infer GOs (eggNOG-mapper) also produces domain annotation via Pfam. We have now performed a parallel analysis based on the gain and loss of **Pfam protein domains** at each of the 11 nodes. This domain-based analysis, which is less reliant on full-length gene homology, provides strong, independent support for our GO-based findings. To infer domain-convergence, we annotated the novel HGs across the 11 terrestrial events with Pfams. The numbers of shared Pfams among all nodes are shown in Figure.2c. A new UpSet diagram highlights the shared domains in at least five terrestrial events (green bars).

Figure.2c Distribution of shared Pfam domains of gene novelty across terrestrialisation. Bars display the number of Pfam domains from novel (in green) and novel core HGs (in blue) shared by at least five terrestrial nodes. Orange bar displays one shared Pfam in fully terrestrial lineages.

The same functional annotations in the GO-based analysis are recovered from Pfam analysis, e.g., ion transport/osmoregulation is recovered by neurotransmitter-gated ion-channel ligand binding domain and neurotransmitter-gated ion-channel transmembrane region, response to stimulus and neuronal functions is recovered by transmembrane receptor (GPCR), detoxification is recovered by Cytochrome P450. This demonstrates that our core conclusions are not an artefact of incomplete gene annotation. We also have completed the Pfams analyses on “Unique Pfams” (Supplementary Figure 7) and semi- and fully terrestrial species comparison (Figure 4b in manuscript), which both correspond with those of GOs. Taken together, the terrestrialisation involved convergent functions of gene novelty are robust to the choice of functional annotation strategy. In addition, for the expanded and contracted HGs, convergence was assessed directly at the HG level; shared enriched GO terms are served as supporting evidence, not as the primary basis for functional inference.

These results have now been added to the manuscript in line 113 for shared Pfams of novel genes, line 136 for Unique Pfams, line 232 for comparison of habitat groups; Figure.2c, Figure.4b, and Supplementary Information 1.5, Supplementary Figure 7.

- **Point 5:** *Line 237: "Most share adaptations between fully terrestrial lineages are found in Arthropoda". This is expected, as it involves parallel transitions in more closely related taxa, suggesting that contingency (i.e., tinkering with what existed before) played a significant role in the transitions to land. In general, what is the impact of phylogenetic relatedness on the conclusions of the work?*

We agree that convergence between closely related arthropod groups is less surprising than between, for example, an arthropod and a vertebrate. Our main argument for convergence is based on the genetic changes or functions shared across 11 different terrestrial lineages, some more related than others. As the referee highlights, in the case of the different terrestrial lineages within arthropods, each of them emerged independently from aquatic ancestors and probably started from a similar genetic toolkit (parallel evolution), consistent with our newly added PCoA and enrichment analysis (lines 230–237). These analyses reveal that fully terrestrial species indeed display a small and streamlined toolkit. This distinction has now been added to the manuscript in line 259 and Supplementary Table 11.

- **Point 6:** *Line 304–309: The colonisation of land by annelids and tetrapods is convergent (i.e. they occurred independently and implied a sea-to-land change). However, the authors suggest that it also involved convergent or similar genetic changes, based on the shared expansion of two gene families: disintegrin and ADAM proteins. Is this sufficient to deem convergence? Also, it is unclear how these two changes would relate to the biological adaptations mentioned in the text (more efficient fertilisation and complex nervous systems). The clitellate nervous system is not more complex than those of marine relatives (e.g. capitellids), and while clitellates evolved more complex fertilisation mechanisms (e.g. clitelum), the early tetrapods (amphibians) retained external fertilisation, and internal fertilisation only evolved later with a strict terrestrialisation. The authors try to establish strong connections between genetic changes that occurred at a node defined by a specific taxonomic sampling to explain a complex adaptive process that likely resulted from many gradual changes before and after that node. While some of this is understandable given the available data, it also presents an unnecessarily biased view towards one evolutionary scenario, especially when the dataset the authors assemble could allow them to explore different hypotheses, as discussed in some of the points above.*

We agree with the referee and have now revised the text to make this distinction clearer. Regarding the ADAM and disintegrin proteins, we agree with the referee, and we have removed them from the text. Tetrapods are difficult to compare to other terrestrial lineages, because amphibians are still heavily reliant on water for key points of their life cycle.

Other minor points...

- *Figure 1: Phoronozoa or Brachiozoa?*
- *Line 25: Kingdom suggesting (space missing)*
- *Line 34: Terrestrial clades also have many developmental adaptations (e.g., encapsulated larvae, brooding, etc.)*
- *Line 37: genotype (typo)*
- *Line 67: were available*
- *Line 128: ENAM should go in italics if referring to the gene.*
- *Line 330: ". We found..." (typo)*
- *Line 384: "events more precisely or language models..." [remove stop after precisely]*
- *Line 412: "we selected some genomes THAT do not perfectly..."*
- *Line 418: Homo sapiens in italics.*

Thank you for spotting these. All typos, errors, and formatting errors have been corrected.

Responses to Referee #2

In this paper the authors approach an important question in the evolutionary history of life on earth, when and the genetic basis of how animals adapted to life on land. We already know that this has happened not so many times, with the successful lineages radiating to produce, alongside the other groups of life, (microbes, plants fungi) the terrestrial ecologies we see today. There have been many studies linking specific genes to specific adaptive traits and asking if these evolve in parallel. However these are generally relatively simple traits wing colour patterns, vertebrate limb reduction/loss, and similar. Terrestrialisation is a different level of adaptation, involving many changes in anatomy, sensory-neural systems, biochemistry and physiology, and their integration. To my knowledge, while there have been studies looking at terrestrialisation in individual lineages, this is the first to take a broad comparative approach across the animal tree and to do so in a systematic way. I think this makes it an important general topic of potential wide interest.

The authors use a pipeline of their making (and previously used and published by them in studies identifying genetic changes at other key evolutionary nodes) to compare the

proteomes of animals to identify gene content changes mapping onto nodes where animals shifted to land. The approach is to define 'homology groups': it has caveats but so do other approaches of correlating genes to adaptations at deep evolutionary nodes, and I think it's the one least affected by caveats. Importantly in my opinion, the approach is both systematic and quantitative, allowing Gene Ontology to be used to compare lineages/nodes. They also use the sequence data to build a time calibrated tree of the lineages involved. Following this is mapping numbers of gains and losses onto the nodes of the tree, including applying a method to determine when there have been expansions or contractions of gene number within homology groups, and whether there are shared patterns to this.

Defining these numbers and the gene groups that comprise them is an important step although I have a concern about the methodology I detail below. The main outcomes beyond this are proposing three episodes when territorialisation happened, and identification of a range of candidate genes that may reflect functions involved in the adaptation under study. As detailed below, I did find the first of these questionable, and the second, while individual cases are believable, is potentially subject to my concern above and has the appearance of being a selectively picked list of intriguing but untested associations.

In summary this is a potentially exciting topic but I do have some significant concerns around methodology, interpretation and the conclusions drawn

We thank Referee #2 for their thoughtful review and for recognizing the originality and systematic nature of our approach.

Point 1: *Line 88 and onwards. I have a methodological concern over the data presented here, how they are interpreted and how this feeds into subsequent figures. As I understand the analysis and data summarized in Figure 1b, the numbers show the gene (homology group) changes that map onto these respective nodes. That's fine, and important to have a consistently applied methodological approach to identifying these. **However surely these numbers will also be dependent on the gap between the base of the node and the next outgroup branch?** I apologise if the authors have explicitly addressed this somewhere in the methods and I missed it, but I think we have to assume that there will be a baseline degree of gene turnover, and hence that a long unbroken branch will generally have higher numbers than a shorter one. **For example if we then look at the data in figure 1b, this seems to fit with woodlice and bdelloid rotifers having high numbers, correlating with the time tree above that shows relatively recent crowns predated by long unbroken branches.** And vice versa for the tetrapod and the other arthropod crowns. I understand that there are inherent limitations in such analyses due to what genomes are available*

and fundamentally to extinction of lineages that would break a branch, but I am struggling to see how we can then read much into the shared changes identified in Figure 3.

This is an important point raised by the referee. We have followed this referee's advice and have calculated the novel gene gains over time, as also indicated in the answer to Point 1a of reviewer #1. For each terrestrialisation node, we divided the number of novel and novel core HGs by branch length (in Myr) on our dated species tree, the resulting evolutionary rates are presented in the table below. As indicated above, we performed a permutation test with these values, showing that the gene gains in terrestrial nodes are significantly higher than in aquatic ones. In addition, the expanded and contracted genes were inferred with CAFE5 shown in Figure 3, whose birth-death model is intrinsically scaled by branch length and does not require any correction. These analyses have now been added to the manuscript in line 98 and Supplementary Table 3.

Novel genes evolutionary rates (rate = novel HGs count / Myr)				
	Type of HG	Novel rates	Novel core rates	Myr
Node1 Bdelloidea	Protostomia	7.528	0.000	572.93
	Lophotrochozoa	0.924	0.000	567.93
	Rotifera	0.630	0.513	419.27
	Bdelloidea	57.243	30.858	129.43
Node2 Clitellata	Pleistoannelida	0.146	0.021	520.12
	Sedentaria	0.480	0.052	485.27
	Capi-Clite	0.245	0.038	448.18
	Clitellata	0.704	0.413	365.31
Node3 Stylommatophora	Heterobranchia	3.705	1.569	174.65
	Panpulmonata	1.614	0.561	144.33
	Pulmonata	0.872	0.570	119.28
	Stylommatophora	3.236	0.946	85.59
Node4 Nematoda	Protostomia	7.528	0.000	572.93
	Ecdysozoa	0.375	0.002	565.11
	Cryptovermes	0.822	0.000	558.2
	Nematoda	5.831	1.241	473.64
Node5 Tardigrada	Ecdysozoa	0.375	0.002	565.11
	Cryptovermes	0.822	0.000	558.2
	Panarthropoda	0.183	0.004	553.25
	Tardigrada	6.640	6.640	187.64
Node6 Onychophora	Cryptovermes	0.822	0.000	558.2
	Panarthropoda	0.183	0.004	553.25
	Lobopodia	0.315	0.002	546.79
	Onychophora	8.402	8.402	546.79

Node7 Arachnida	Arthropoda	1.341	0.007	536.83
	Chelicerata	0.023	0.000	519.41
	Euchelicerata	0.301	0.029	478.76
	Arachnida	0.098	0.007	460.98
Node8 Myriapoda	Lobopodia	0.315	0.002	546.79
	Arthropoda	1.341	0.007	536.83
	Mandibulata	0.701	0.000	527.51
	Myriapoda	0.171	0.064	469
Node9 Armadillidium	Communostraca	0.091	0.031	419.87
	Malacostraca	2.043	1.162	280.52
	Peracarida	0.041	0.041	219.1
	Armadillidium	6.203	6.203	219.1
Node10 Hexapoda	Pancrustacea	0.224	0.008	517.34
	Altocrustacea	0.758	0.006	504.03
	Allotriocarida	0.159	0.004	482.77
	Hexapoda	0.115	0.019	462.61
Node11 Tetrapoda	Osteichthyans	0.740	0.005	431.33
	Sarcopterygii	0.104	0.007	404.02
	Dipnotetrapodomorpha	0.214	0.003	387.36
	Tetrapoda	0.780	0.009	344.9

- Point 2:** *Figure 3a shows shared gene groups between the various combinations of terrestrialised lineages. It took a bit of time to work out from the text that the authors had used CAFE5 to build the lists of expansion/contraction at each node, this wasn't mentioned till the discussion. Understanding this after some effort, I do appreciate that this software is more sophisticated than just counting, and does model rates in some form. But, if we have nodes that have accumulated lots of change in part due to long unbroken branches preceding them, are we not more likely to see overlap in expansion/contraction of gene groups as a consequence of the higher numbers? See also my comment regarding line 152 below. I am not arguing that changes linked to terrestrialisation won't be captured here as well, I am pretty sure they will, but I am concerned the quantitative approach taken by the authors in this study, which is admirable as I say elsewhere, is compromised by not disentangling these. Does this not then feed in to undermining the GO distribution analyses as well?*

We apologise and we have now made clearer that we use CAFE5 to infer expansions and contractions, in manuscript lines 85 and 100. As indicated above, and correctly identified by the referee, CAFE5 likelihood tests use a birth-death model, which is intrinsically scaled by branch

length. Moreover, our new analyses raised in the previous point show that gene gains are also larger even accounting for divergence time.

- **Point 3:** *Line 152: this relates to my comment above. The gene types identified here make sense in terms of adaptation, but I think some of them are well known as families that undergo relatively rapid changes in copy number in many lineages, most unrelated to terrestrialisation. For example the variable numbers of glutathione s-transferases in insects and cephalopods, of olfactory receptors in many lineages. How can we conclude the cross-lineage similarities observed here aren't a consequence of this general high lability and not the specific adaptation under study here? **If they are genes likely to change in copy number anyway, wont they be more likely to show as shared between nodes that have long preceding branches?***

This is a critical point about a potential major confounding factor in comparative genomics. We agree that families like GSTs and olfactory receptors are known to be labile. However, the strength in these analyses is that these genes only expand in terrestrial lineages (in the GST case, in disparate groups such as bdelloid rotifers, tardigrades, nematodes, and tetrapods) and not in aquatic ones, and convergently (not just in one lineage). Our new GOs permutation test (described in our response to Referee #1) also demonstrates that the convergent functions derived from the gene gains across 11 independent events are highly unlikely to be a product of chance alone.

- **Point 4:** *Line 200: here the authors divide the lineages into semi and fully terrestrial. I'm in two minds about this. There are obviously differences in this 'axis' of life history, and quantifying it isn't possible so qualitative subdivision is justifiable. **But might another axis/subdivision make more biological sense, eg between miniature interstitial animals and others? Ones with big brains/sense organs and others? Or does this confuse terrestrialisation with adaptation to micro versus macro life? I don't have a strong view, but do think the authors need to explain why the subdivision they have chosen is the right one.***

This is an exciting comment. We have to start by saying that we were not able to find a consensus definition of types of terrestrial animals, so we based our definition on animals' reliance on water. We have clarified this in the manuscript line 220 and 362.

We agree that body size, nervous system, sensory systems, and other adaptations are definitely related to terrestrialisation; in the manuscript, we argue that our findings support the further evolution of the latter two in this transition. Body size is definitely large in all the fully terrestrial

animals (arthropods, tetrapods, snails, slugs), while semi-terrestrial comprise mostly small organisms like bdelloid rotifers, nematodes, tardigrades. However, our dissection of the functions driving the PCoA plot and the shared novel gene GOs between fully terrestrial vs. semi-terrestrial do not point to body size being a factor, and the convergence level of the (large) fully terrestrial animals does not point towards body size being a factor: we actually find no shared functions and very few enriched functions (Supplementary Table 11) at all amongst fully terrestrial animals, except within the different terrestrial arthropod lineages (lines 257 to 266).

- **Point 5:** *Line 274 onwards. This section of the results concerns the relative timing of the terrestrialisation events. I have no concerns about the time tree analysis shown in figure 1, the methods look robust. I am less persuaded about the definition of three episodes. Inevitably for this sort of analysis the probability windows for when these changes happened are quite large. Overlapping terrestrialisations could be fully congruent. Or they could be separated by many millions of years. This means they could be emerging in similar ecological and climatic environments, or not. How then are we to interpret a small number of shared gene changes, like for example the ADAM protease expansions in tetrapods and clitellates, when we also think about their separation into semi versus fully terrestrial earlier on in the paper, and their adaptation into (I think, I'm not an expert in the paleo evidence here) burrowing versus mobile predatory life histories? I am struggling to see how we move from an interesting observation to deeper explanatory understanding especially given the comments I made above. I don't think the manuscript makes this leap.*

We agree that the confidence intervals on our date estimates are large, and the term "episodes" may have implied overly discrete events. We have revised this section and now refer to them as three "**temporal windows**" of terrestrialisation and clarified that they might not be overlapping events, which may be separated by millions of years. Following a comment from Referee #1 we have also removed the mention of convergence between tetrapods and clitellates.

Suggested Improvements (Abstract, Intro, Discussion):

Abstract

- *In lines 14-15 there is a claim made that this approach allows one to decipher the roles of contingency and convergence in adaptation, however I don't think this is actually discussed in the manuscript*

Agreed, we have now added a new closing paragraph in discussion: “Some genomic adaptations linked to terrestriation show clear convergent patterns, suggesting that certain molecular responses to life on land are broadly predictable and have evolved independently in multiple animal lineages replaying the tape of life. Yet, convergence is only part of the story. Each terrestrial lineage also displays its own contingent adaptations, shaped by its unique evolutionary history, genomic background, and ecological context. Even when facing similar challenges, different lineages often arrive at distinct molecular solutions, reflecting their ancestral constraints and trajectories. Terrestriation, therefore, illustrates the interplay between convergence and contingency, highlighting both the repeatability and the uniqueness of evolutionary innovation”.

- *Late on, the abstract mentions regeneration. This is one small thing mentioned at one point in the paper and as detailed below I am sceptical as to its significance anyway, I don't think it should be in the abstract.*

We believe our findings about regeneration are novel and strong enough to be featured in the Abstract (please, see answer to Point 2 of referee #1 and to this referee below).

Introduction

- *Line 29: a small thing, but calling it 'one of the most iconic episodes' is counter to previous views and this paper's conclusions that territorialisation happened at multiple times spread over the last ~500 million years.*

Thanks, this has been changed to “one of the most significant transitions”.

- *Line 33: similarly, I don't think we can claim these lineages faced the same ecological challenges. Animals adapting to terrestrial live at different times faced hugely different ecologies dependent on what had already territorialised and diversified previously.*

Thanks, this has been changed to “similar physiological and environmental challenges”.

- *Line 47: Delete vast here. As well as unnecessary, in this age 147 genomes isn't vast!*

The word “vast” has been removed.

- *Line 107: this paragraph starts by stating something that hasn't been shown yet by the study. Its not wrong because they come on to mention this but it left me floundering to follow the reasoning for a bit till I read on a while.*

Thanks, this sentence has been moved to the end of the paragraph.

- *Line 127: The use here of a couple of gene families subsequently then lost in a lineage returning to the water wasn't convincing to me. Seems cherry picked, there must be losses all over the tree on other nodes, why just pick on these two? Suggest either bolster this with a quantitative analysis or remove it.*

Thanks, these examples have been removed.

- *Line 182: as for my comment on gene loss in cetaceans, two GO terms related to regeneration is not compelling. I could maybe be persuaded if there was good evidence that this homology group was involved in regeneration across multiple lineages, but as the authors discuss GO annotation is heavily skewed to a couple of lineages, and the genes themselves look like general tissue integrity and cellular system genes that have picked up regeneration GO annotation somewhere. I have a parallel thought on the chlorophyllase point below: I thought that fossil evidence was pretty clear that the Devonian aquatic ancestors of tetrapods were predators not herbivores? Happy to be corrected about that though.*

Thanks for this comment. In the manuscript we state that there are two gene families convergently lost, although we didn't clarify in how many lineages. As we indicated in our answer to Point 2 of reviewer #1, we believe convergent evolution is a great approach to find shared patterns in multiple transitions. We believe it is striking that these two genes related to regeneration have been widely lost. Dbl gene family was lost in **8** out of 11 terrestrial events (retained only in bdelloids, Stylommatophora and tetrapods), and the pleckstrin-domain gene family was lost in **7** out of 11 terrestrial events (retained only in bdelloids, Stylommatophora, myriapods and tetrapods; Supplementary Table 9). For reference, we do not find gene families convergently expanded or contracted in more than four nodes (Figure 3 in the main manuscript). Our results therefore suggest a tendency, with repeated loss of regeneration components. In addition, the annotations for these lost gene families were obtained directly from **UniProt entries**, rather than inferred from GO terms. We have reworded the title of that section to better illustrate our findings.

About the vegetal diet in tetrapods, comparative genomics analyses capture the genome content in the last common ancestor of the extant members of a lineage rather than the first ancestor of the clade, as we have no access to genomes of the extinct taxa. Thus, some discrepancies may happen between the fossil record and the last common ancestor of the group.

- *Line 291: do the authors mean land environments here? Or is a landing environment something else, where they first came onto land maybe?*

Yes, that was a mistake, it has been corrected now. Thanks!

Responses to Referee #3

This manuscript employs 154 complete genomes representative of the breadth of animal phylogeny in an analysis that concludes “there is strong evidence of convergent genome evolution...” with respect to territorialization events over the course of animal evolution. While the manuscript is attempting to address a broadly interesting evolutionary question, several aspects about this work render it problematic. Without digressing into excessive detail, some of the major concerns are mentioned here. However, before mentioning concerns about assumptions, analysis, or conclusions, the overall grammatical quality of the paper should be noted. This paper clearly was not well vetted before submission given the numerous glaring errors and typos.

We thank Referee #3 for their critical reading and for pushing us to adhere to the highest methodological and presentational standards. We apologize for the grammatical errors in the initial submission and have now thoroughly proofread the entire manuscript.

- **Point 1:** *The main concern is that the analyses undertaken in this work is not up to current molecular evolutionary or phylogenetic standards. Basically, **the work builds a phylogenetic topology from the data at hand, uses a BLAST approach to identify genes or homology groups, and used GO terms for functional analyses.***

We appreciate the reviewer's concern for methodological rigor. We apologize if there is any confusion, as we have not built any phylogenetic tree in this study. The tree depicted in Figure 1 is a guide tree based on the literature (e.g., Laumer et al 2019⁴), for which branch lengths (but not the topology) were determined using BUSCO genes. This is a typical approach to infer timetrees and the scaffold for comparative genomics analyses (e.g., our pipeline, CAFÉ5). We have rewritten the material and methods section to make this clearer. Regarding the identification of homology groups, we used **OrthoFinder**, a widely used, graph-based method

using Markov clustering that is more accurate than simple reciprocal-best-BLAST and it is a staple in the field with thousands of citations. Finally, we have complemented the GO analyses with Pfam domains, at the request of the other reviewers; Pfam annotation supports our findings and validates the GO results.

- **Point 2:** More *discerning methods would include some type of phylogenetic independent contrasts approach augmented with individual gene trees for the HG to assess gene family (or HG) expansion and contraction. Just employing GO terms over such a broad swath of the animal kingdom is problematic as most of the function is drawn from a few select models (as noted on line 364). More attention to which finding were significant would have been appreciated (example line 150).*

While we agree that individual gene tree analyses are powerful, they are computationally prohibitive for 480,000 HGs across 154 genomes in our dataset, with some of the HG comprising thousands and thousands of sequences. Similarly, to use these trees to perform phylogenetic independent contrasts of the functional annotations, we would need to infer the GO/Pfam annotations for each of the 154 species, which is not realistic. Our approach uses state-of-the-art methods which are staples of the field, combining robust orthology inference with ancestral state reconstruction and rate analyses (CAFE5), representing the current standard for large-scale comparative genomic analyses of gene content evolution using approaches that gather thousands of citations.

As explained above, we have now complemented the GO analyses with Pfam domains, which add another layer to our functional annotation, and performed statistical tests on both quantitative and qualitative innovations during terrestrialisation.

Other issues:

- **Point 3:** *There is no indication given of node support on the phylogeny. This is an issue as some branches vary from the current best estimates of animal phylogeny (e.g., byzoan placement, sponge placement [this is no longer controversial]). It is ok that these placements are not recovered, but the presentation needs to be clear on this issue and present analyses that will allow readers to assess the tree.*

As indicated above, we would like to clarify that we did not reconstruct any phylogenetic tree as part of this study, thus there are no node supports. Instead, we deduced a guide tree from well-established published phylogenies to define the relationships among taxa and to map the

evolutionary nodes corresponding to transitions to land. Importantly, the problematic taxa like sponges, ctenophores, Xenacoelomorpha, and the position of bryozoans are far removed from the terrestrial lineages of interest.

- **Point 4:** *Lost HG are defined as “HG lost in the all species with a node, while present in the sister groups or other species in outgroup.” First there is only one sister group to a give node (assuming the node is resolved). Second this definition also defines HGs that are autapomorphies in the sister lineage or an outgroup lineage. That is, it contains more than “lost HGs” and autapomorphic HGs in sister taxa likely skew the calculations if done as presented.*

We believe there is a confusion here, in *Material and Methods* line 458 we defined lost genes as “Lost HGs: HGs lost in all species within a node, while present in the sister groups and other species in outgroup”. Therefore, they are not autapomorphies of the sister group to the terrestrial lineage.

- **Point 5:** *In some instances of text appear contradictory. For example, the Abstract states, “there is strong evidence of genome evolution” but in the Introduction states, “...our results reveal that independent terrestrial events were driven by the emergence of similar biological function, although semi- and fully terrestrial lineages exhibit different patterns of genomic adaptation...”.*

We do not believe these statements are contradictory. We identify genome evolution (changes in gene repertoire, gains and reductions), but these changes lead to the same or similar biological functions and adaptations, thus convergent evolution. Some biological adaptations to terrestrial environments are shared by most terrestrial lineages, some are only shared by semi-terrestrial, and some are lineage specific (the latter ones being non-convergent).

- **Point 6:** *Better justification is needed as to why some clades were a focus (e.g. rotifers) and others were not (molluscs).*

We believe that there are no biases toward specific clades in the manuscript. When discussing intrinsic changes to each lineage (line 270), rotifers and molluscs are equally discussed (lines 271 to 272). All clades are discussed throughout the manuscript, except in the last section on phyla-specific evolution. There we focus on arthropods and tetrapods, for the sake of space and because they are major clades of high interest to a general audience; but as indicated (line 269) we discuss every single lineage in Supplementary Information 1.4.

- **Point 7:** *It is not clear if the gene families focused on were considered ahead of time or were post hoc. If the latter, there should be some discussion of how many gene families showed*

change. With enough comparisons some HGs will show significant differences. It is noted that a correction for multiple analyses was included in the GO term analyses.

We apologize if this was not clear, we have not selected any gene family a priori, all the gene families analysed are result of the comparative genomic analyses. We rephrased this in the paper to make it clearer (line 121). We analysed all HGs for gains/losses with our pipeline, and expansions/contractions with statistical significance determined by CAFE5, now we have also included permutation tests and Pfam analyses. We then discussed specific gene families with known biological relevance *a posteriori* to illustrate these statistically significant, large-scale patterns.

- **Point 8:** *There is a good bit of jargon and limited discussion in some cases to back up ideas. As a result, good chunks of the discussion seem ad hoc. For example lines 126-129 describing APOA2 and ENAM being involved in secondary transitions. Why these genes? This part of discussion seems a bit rambling. A better pretext discussing the genes and gene functions that would be needed, or lost, for terrestrialization should be given and then the work should specifically test for those genes or HGs.*

We have revised the discussion significantly to improve its flow and provide better context for all examples, ensuring that claims are clearly linked back to the evidence presented in our results. The section on APOA2/ENAM has been removed based on the comments of this and another referee.

Reference

- 1 Bely, A. E. Distribution of segment regeneration ability in the Annelida. *Integr. Comp. Biol.* **46**, 508-18 (2006). <https://doi.org/10.1093/icb/icj051>
- 2 Bely, A. E., Zattara, E. E. & Sikes, J. M. Regeneration in spiralian: evolutionary patterns and developmental processes. *Int J Dev Biol* **58**, 623-34 (2014). <https://doi.org/10.1387/ijdb.140142ab>
- 3 Ozpolat, B. D. & Bely, A. E. Developmental and molecular biology of annelid regeneration: a comparative review of recent studies. *Curr Opin Genet Dev* **40**, 144-153 (2016). <https://doi.org/10.1016/j.gde.2016.07.010>
- 4 Laumer, C. E. *et al.* Revisiting metazoan phylogeny with genomic sampling of all phyla. *Proc. Biol. Sci.* **286**, 20190831 (2019). <https://doi.org/10.1098/rspb.2019.0831>

Response to Referee #1:

In this revised version of the manuscript entitled "Convergent genome evolution shaped the emergence of terrestrial animals", Wei and co-authors convincingly address the previously raised points. The statistical analyses of the changes in gene family composition are smart and strengthen the original observations. Likewise, the inclusion of gene functional enrichments based on PFAM annotation supports the original gene-based approaches, and the text is now generally more balanced and accurate. Overall, this is a fascinating study that addresses one of the crucial outstanding questions in animal evolution and will open new avenues for investigating animal adaptation to land at multiple levels, from palaeobiology to genetics, development, and cell biology.

I do not have further experimental concerns, but there are two main points the author should consider.

We appreciate the comments and support of this referee and their support of our analyses.

- I still find the connection between terrestriation and loss of regeneration poorly supported. While it might be true that terrestrial lineages exhibit, at least ancestrally, poor regeneration capacities, I do not see how the convergent loss of two gene families (dbl and plekstrin-related) can be proof of a genetic basis of that regeneration reduction. These are broad gene families that are involved in many biological processes, and the references provided to support their role in regeneration are limited to vertebrates and cell type regeneration, rather than organ/whole body regeneration. The interpretation as a "shift away from a more generalised aquatic larval regeneration" (line 437) is also unclear and based on a manuscript (REF 49) that presents a very unique hypothesis on the nature of regeneration (which is an ongoing debate in the field). Indeed, there is no such thing as "aquatic larval regeneration" as many aquatic systems with strong regenerative capacities do not have a larval stage (e.g., acoels). It is thus widely speculative that the loss of these two families is causally related to poor regenerative abilities across distantly phylogenetically related animal phyla, especially when the terrestrial lineages lacking those gene families likely share many other traits that could also be linked to the potential biological roles of these genes. Nevertheless, if the authors want to keep this interpretation, I suggest to remove it from the abstract (and use those few words to highlight other better supported points of the manuscript, such as the connection of the three windows of terrestriation with different geological/ecological conditions) and tone down/qualify the section about loss of regeneration (and probably remove the associated discussion), indicating that the function of these gene families in most terrestrial lineages is unknown and that their loss might be either a striking coincidence (ie., non-adaptive and by drift) or linked to other traits those lineages might share. To me, the convergent loss of chlorophyllase in terrestrial lineages seems more interesting as it might be related to a shift from an algal-rich diet to

one based on land plants (again, speculative, but perhaps there is something in there that makes more sense than a vague connection of two broad gene families to neuronal/muscle cell regeneration in vertebrates).

We thank the reviewer for this suggestion. We have removed the regeneration related description from the abstract, and expanded there the link between the three timescale windows with specific ecological conditions (respectively, lines 20 and 22). Also, we have toned down the section about lost genes related to regeneration in both results section (used to be between the current line numbers 167-168) and discussion section (current line 301).

- The text still needs some polishing and grammar correction, both in the main manuscript and in the supplementary technical notes (there are many examples, but to pick a few: all figures are called as Figure. [with a dot at the end]; In line 111, the paragraph starts as "To infer this functional convergence", but that is a legacy of the previous version because the convergence has not been mentioned yet. Additionally, the text mixes British and American spelling, which should also be avoided. The Discussion is redundant at times and could be streamlined into two or three paragraphs that reflect on the key findings and limitations of the study.

We thank the reviewer for spotting these. We have now thoroughly proofread the entire manuscript and supplementary information and corrected the grammatical errors. We have also removed the redundant content in the discussion and streamlined it into three paragraphs.

Minor points:

- Lines 477–480: How does this study support the phylogenetic position of Xiphosura and Arachnida? There is no phylogenetic reconstruction in the study, and the phylogenetic position of any group should not be based on patterns of gene gain/loss.

We thank the reviewer for this critical observation. We have now rephrased these description in discussion: "We followed recent studies that place Xiphosura as the sister group to Arachnida, implying a single origin of terrestrialisation in arachnids, whereas some placements propose that Xiphosura may be nested within arachnids, which would suggest an alternative scenario." (line 336)

- Finally, the GitHub repository might need to be updated with the code for the new analyses included in the manuscript.

Thanks, the code for the new analyses has been updated in the GitHub repository.

Referee #1 (Remarks on code availability):

The repository might need update with the latest analyses and scripts, but it is generally comprehensive and should ensure reproducibility.

Thanks, the code for the new analyses has been updated in the GitHub repository.

Referee #2 (Remarks to the Author):

The authors have addressed the important methodological concerns I raised, especially regarding controlling for rates of gene turnover. I still find the emphasis on regeneration suspect, especially its inclusion in the abstract, when this is based on such a small number of genes with broad cellular functions. I think this could well be due to bias in how GO terms are attributed to genes. However, its not wrong for the authors to examine this and discuss, its just the emphasis and I would prefer it to be cut from the abstract.

We thank this referee for their support and positive comments. The point about regeneration is very fair. We have removed the regeneration related description from abstract, and have toned down the section about lost genes related to regeneration in both results section (used to be between the current line numbers 167-168) and discussion section (current line 301).